# Expressiveness of Parametrized Distributions over DAGs for Causal Discovery

**Simon Rittel**                                                              *simon.rittel@lmu.de*
*Department of Statistics, LMU Munich, Germany*
*Munich Center for Machine Learning, Germany*
*UniVie Doctoral School Computer Science, Austria*

**Sebastian Tschiatschek**                            *sebastian.tschiatschek@univie.ac.at*
*Faculty of Computer Science, University of Vienna, Austria*
*Research Network Data Science, University of Vienna, Austria*

**Reviewed on OpenReview:** *https://openreview.net/forum?id=UsJOH6VJRl*

## Abstract

Bayesian approaches to causal discovery can—in principle—quantify uncertainty in the prediction of the underlying causal structure, typically modeled by a directed acyclic graph (DAG). Various semi-implicit models for parametrized distributions over DAGs have been proposed, but their limitations have not been studied thoroughly. In this work, we focus on the expressiveness of parametrized distributions over DAGs in the context of causal structure learning and show several limitations of candidate models in a theoretical analysis and validate them empirically in relevant supervised settings. To overcome these limitations, we propose using mixture models of the considered distributions over DAGs, demonstrating improved flexibility and performance.

## 1 Introduction

Causal discovery, also known as *causal structure learning* (CSL), is the task of uncovering cause and effect relations among modeled variables based on observed data (Glymour et al., 2019; Squires & Uhler, 2022). The inferred structures govern the translation of a causal estimand into a statistical one that can be measured from data and, hence, provide a basis for causal inference (Pearl, 2009; Lundberg et al., 2021). Errors in causal discovery that occur, e.g., due to limited data availability as well as modeling choices, can thus imply different statistical estimands and bias the analysis of causal queries. This dependence highlights the importance of quantifying the uncertainty of inferred causal structures for causal inference.

To this end, Bayesian CSL goes beyond identifying only a single, potentially incorrect, causal structure and rather models the uncertainty over the true causal structure. It incorporates prior knowledge in the form of a prior distribution and—in principle—enables the computation of a posterior distribution over possible causal structures when new evidence is presented. Typical assumptions for causal discovery are causal sufficiency, the absence of selection bias, and acyclicity of the causal graph, allowing to represent the causal structure by a *directed acyclic graph* (DAG). Given asymptotic identifiability of the causal graph, two sources of error for its prediction can be distinguished: (i) The *approximation error* results from the finite amount of evidence for the estimation of the possibly nonlinear relations among the observed variables and decreases with increasing data size; (ii) the *model error* results from the choices for the functional relationships among variables and the distribution over the causal graph and cannot be overcome with more data.

In our work, we focus on the model error and investigate the importance of the model choice for the distribution over causal structures. This is particularly important as novel Bayesian CSL algorithms introduced in recent works often differ in multiple model choices and only rarely identify for all of them which key detail is responsible for the reported increase in performance in comparison to competing models. While a branch

of research focuses exclusively on the functional relationships and parametric forms that allow identification of cause and effect pairs (Shimizu et al., 2006; Hoyer et al., 2008; Zhang & Hyvärinen, 2009; Loh & Bühlmann, 2014; Immer et al., 2023), distributions over causal graphs, e.g., DAGs, have so far received only little attention.

**Contributions**  In our work, we investigate and compare the expressiveness of distributions over DAGs used in recent Bayesian CSL algorithms (Cundy et al., 2021; Lorch et al., 2021; Charpentier et al., 2022; Deleu et al., 2022; Rittel & Tschiatschek, 2023; Annadani et al., 2023; Toth et al., 2024). We highlight their limitations in assigning equal probabilities to graphs of the same Markov equivalence class (Example 1) as well as capturing dependencies between edges in the graphs (Example 2) and compare their ability to match synthetic graph distributions. In addition, we provide experimental and theoretical evidence that the probabilistic models proposed in Rittel & Tschiatschek (2023); Deleu et al. (2022); Toth et al. (2024) and mixtures of them are more expressive than simpler particle distributions. We believe that our work will be helpful to researchers and practitioners alike by demonstrating shortcomings of recently proposed distributions over DAGs that can limit the applicability of Bayesian CSL and proposing mixture models as an effective countermeasure. In addition, we provide a recipe for efficiently approximating the probability mass function of a given graph, a directed path, or a subgraph induced by a generative model for the causal DAG by marginalization over an auxiliary random structure using importance samples.

**Structure of the paper**  The remainder of this paper is structures as follows. In section 2, we introduce our notation and background on causal discovery. We proceed in  section 3 with the presentation of the considered distributions over DAGs and discuss their theoretical limitations. In section 4, we outline an efficient method to evaluate generative models based on importance sampling and derive analytically the minimal statistical divergences for particle distributions with $K$ graphs. Finally, we apply both findings for supervised learning of different target distributions in section 5 and conclude our paper in  section 6.

## 2   Preliminaries

**Probability and random variables**  We denote scalar random variables by lower case symbols, e.g., y, random vectors and random matrices by upper case bold symbols, e.g., $\mathbf{Y}$. Single elements of a random vector or matrix are written as Y. The probability of a discrete random variable y taking the value $y$ is expressed by the probability mass function $p_y(y) := \mathbb{P}(y = y)$. When clear from the context, we omit the random variable in the subscript of a probability mass function. For random variables with parameterized probability mass functions with parameters $\boldsymbol{\theta}$, we write $p_{\boldsymbol{\theta}}(y)$. With a slight abuse of terminology, we refer to a distribution $P_y$ by its induced probability mass function $p_y(y) = P_y(\{y\})$. To distinguish approximations from the true target distribution, we denote model distributions by $q_y(y) = Q_y(\{y\})$ or simply $q_{\boldsymbol{\theta}}(y)$.

**Functional causal models**  A *functional causal model* (FCM) is defined as a triple $\mathcal{M}_{\mathbf{X}} := \{\mathbf{X}, (\boldsymbol{\epsilon}, P_{\boldsymbol{\epsilon}}), \boldsymbol{f}\}$ consisting of a set of endogenous random variables $\mathbf{X}$, a set of exogenous noise variables $\boldsymbol{\epsilon}$ with joint probability distribution $P_{\boldsymbol{\epsilon}}$ and a set of deterministic functions $\boldsymbol{f}$, all three indexed by $[D] := \{1, ..., D\}$. Each endogenous variable $X_d$ of the model is generated as a function of a subset of the endogenous variables $\mathbf{X}$ and its exogenous noise $\varepsilon_d$, i.e., $X_d := f_d(\mathbf{X}, \varepsilon_d)$. The distribution of $X_d$ is implicitly defined as the pushforward measure of $P_{\boldsymbol{\epsilon}}$ through the *causal mechanism* $f_d$. The structure induced by the direct functional dependencies is often restricted to be acyclic such that it can be represented by a *directed acyclic graph* DAG or—equivalently—its adjacency matrix $\boldsymbol{G} \in \mathcal{G} \subset \mathcal{A} := \{0, 1\}^{D \times D}$ with a one-to-one correspondence between random variables and nodes. Let $\sim d$ denote the index set $[D] \setminus \{d\}$, then the $d$-th column of $\boldsymbol{G}$ encodes the parents $\mathrm{Pa}_{\boldsymbol{G}}(X_d) \subseteq \mathbf{X}_{\sim d}$ of a node/random variable $X_d$, i.e., the subset of $\mathbf{X}_{\sim d}$ that has a direct influence on $X_d$ via $f_d$. Ancestors of a random variable, $\mathrm{An}_{\boldsymbol{G}}(X_d)$, have a directed path to $X_d$ in the causal graph $\boldsymbol{G}$. If they are not themselves parents of $X_d$, then their causal effect on $X_d$ is mediated by at least one parent of $X_d$. Children $\mathrm{Ch}_{\boldsymbol{G}}(X_d)$ and descendants $\mathrm{De}_{\boldsymbol{G}}(X_d)$ are causally affected by changes of the value of the corresponding node $X_d$ and are defined as inverse relations to parents and ancestors.

**Causal structure learning**  The objective of *causal structure learning* (CSL) is to infer the underlying causal graph $\boldsymbol{G}$ from observed random variables $\mathbf{X}$ that encodes the causal effects implied by the FCM

$q(\mathcal{D}|\boldsymbol{G})$
$p(\mathcal{D}|\boldsymbol{G})$     $q(\boldsymbol{G})$     $q(\boldsymbol{G}|\mathcal{D})$
$p(\boldsymbol{G})$     $p(\boldsymbol{G}|\mathcal{D})$

(a)       (b)       (c)

Figure 1: Modeling errors in Bayesian causal discovery due to approximations $q$ of *(a)* the marginal likelihood function and the model for *(b)* the prior and *(c)* the posterior over the causal DAG $\boldsymbol{G}$. For different graphs they either imply underestimation or overestimation.

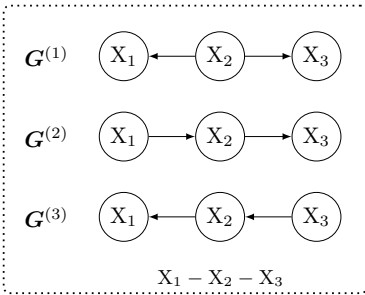

Figure 2: Graphs of the MEC from Example 1 and its CPDAG (below).

$\mathcal{M}_{\mathbf{X}}$. Throughout this work, we assume causal sufficiency, i.e., all endogenous variables $\mathbf{X}$ are observable, and that the exogenous noise variables $\boldsymbol{\epsilon}$ are mutually independent. This implies that all dependencies and independencies between the observed values of the random variables $\mathbf{X}$ result from their causal effects over the functions $\boldsymbol{f}$ and not from some latent common causes, i.e., an unobserved shared ancestors of them. In addition, we assume that all samples in the data set $\mathcal{D} := \{\boldsymbol{X}^{(n)}\}_{n=1}^{N}$ are generated i.i.d. from the FCM $\mathcal{M}_{\mathbf{X}}$ without any selection bias for the generated samples, e.g., there is no conditioning on unobserved confounders.

Assuming faithfulness, the *Markov equivalence class* (MEC) of a causal graph $\boldsymbol{G}$ can be consistently identified from $\mathcal{D}$ using *conditional independence* (CI) tests (Spirtes et al., 2000). This equivalence class contains all DAGs that entail the same observed independence relations and can be compactly represented by a *completely partially directed acyclic graph* (CPDAG), an acyclic mixed graph that has the same adjacencies, $X_i - X_j$, and unshielded colliders, $X_i \rightarrow X_k \leftarrow X_j$, as the underlying true graph $\boldsymbol{G}$, but with some of its edges remaining undirected. While CI tests can be easily parallelized and their required overall number can be sequentially restricted by the individual test results as in the PC algorithm (Spirtes et al., 2000), the combination of the uncertainty attached to each test is non-trivial. Moreover, CI testing for continuous random variables lacks statistical power against alternatives and suffers from the curse of dimensionality (Shah & Peters, 2020).

**Bayesian causal structure learning** An alternative to constraint-based CSL with CI tests are score-based algorithms that optimize a scalar quantity that is typically derived from the likelihood of the observed data $p_{\boldsymbol{\Theta}}(\mathcal{D}|\boldsymbol{G})$. Bayesian causal structure learning is also based on the likelihood but models the full data-generating process for the joint distribution:

$$p(\mathcal{D}, \boldsymbol{\Theta}, \boldsymbol{G}) = p(\mathcal{D}|\boldsymbol{\Theta}, \boldsymbol{G})p(\boldsymbol{\Theta}, \boldsymbol{G}) = \prod_{n=1}^{N} p(\boldsymbol{X}^{(n)}|\boldsymbol{\Theta}, \boldsymbol{G})p(\boldsymbol{\Theta}|\boldsymbol{G})p(\boldsymbol{G}) \,, \tag{1}$$

that includes the uncertainty over model parameters $\boldsymbol{\Theta}$ as well as a prior distribution over plausible causal graphs $p(\boldsymbol{G})$. The posterior distribution over the causal $p(\boldsymbol{G}|\mathcal{D})$ that quantifies the uncertainty over the true causal graph is then obtained by Bayes' formula:

$$p(\boldsymbol{G}|\mathcal{D}) = \underbrace{\int p(\mathcal{D}|\boldsymbol{\Theta}, \boldsymbol{G})p(\boldsymbol{\Theta}|\boldsymbol{G}) \,\mathrm{d}\boldsymbol{\Theta}}_{=p(\mathcal{D}|\boldsymbol{G})} \frac{p(\boldsymbol{G})}{p(\mathcal{D})} \,, \tag{2}$$

where the so-called evidence $p(\mathcal{D})$ results from marginalization of Eq. (1) over $\boldsymbol{\Theta}$ and $\boldsymbol{G}$. Averaging the full likelihood $p(\mathcal{D}|\boldsymbol{\Theta}, \boldsymbol{G})$ over the (conditional) prior distribution of model parameters yields the marginal likelihood denoted by $p(\mathcal{D}|\boldsymbol{G})$. In contrast to a plain maximum likelihood estimate $p_{\boldsymbol{\Theta}_{\mathrm{ML}}}(\mathcal{D}|\boldsymbol{G})$, it avoids overfitting to the noise of the data (Koller & Friedman, 2009) and motivates the stochastic modeling of $\boldsymbol{\Theta}$. In general, computing the evidence involves marginalization over all DAGs, $\boldsymbol{G} \in \mathcal{G}$, and parameters of the likelihood function, $\boldsymbol{\Theta} \in \mathbb{R}^p$, and is intractable. Therefore, approximate methods for Bayesian inference are

needed. In the following, we focus on variational inference where the joint distribution $p(\boldsymbol{G}, \mathcal{D})$ is approximated by a generative model $q(\mathcal{D}|\boldsymbol{G})q(\boldsymbol{G})$, and a variational family of distributions $\mathcal{Q} := \{q_{\boldsymbol{\phi}}(\boldsymbol{G}|\mathcal{D}) \,|\, \boldsymbol{\phi} \in \mathbb{R}^p\}$ is specified for the intractable posterior $p(\boldsymbol{G}|\mathcal{D})$. Both components involve parametric distributions over DAGs.

**Bayesian model error**  Given a fixed data set $\mathcal{D}$, deviations of a modeled posterior $q(\boldsymbol{G}|\mathcal{D})$ to the true posterior $p(\boldsymbol{G}|\mathcal{D})$ can be backtracked to three different model errors as depicted in Figure 1: *(a)* The modeled marginal likelihood $q(\mathcal{D}|\boldsymbol{G})$ may induce a bias, with the particular case of the maximum likelihood estimate leading to an overconfident prediction. Throughout this work, we assume a flexible model for the marginal likelihood, and focus on the role of the distribution over DAGs instead. *(b)* The model for the prior distribution $q(\boldsymbol{G})$ directly constrains the belief that domain experts can express over the causal structure, but the subjective bias vanishes in the asymptotic limit as long as it assigns each DAG some positive probability mass. *(c)* Lastly, the true posterior may not be in the variational family $\mathcal{Q}$, i.e., it cannot be expressed by $q(\boldsymbol{G}|\mathcal{D})$.

## 3 Distributions over DAGs

In this section we discuss different possibilities for representing distributions over DAGs. While the number of DAGs for a given number of variables $D$ is finite, it is super-exponential in $D$ (OEIS Foundation Inc., 2023; Stanley, 1973). Consequently, using categorical distributions that allow to distribute the probability mass arbitrarily among all graphs quickly becomes infeasible, even for small $D$. However, for many applications, it is not necessary to specify arbitrary probabilities for all DAGs, i.e., a smaller degree of freedom may suffice. For causal discovery, the posterior distribution is expected to concentrate on graphs that are *similar* to the ground truth graph according to which the observed data was generated. This motivates the development of probabilistic models for graphs that are flexible enough to model any possible DAG with $D$ nodes and to additionally distribute the probability mass among some candidate graphs but require substantially fewer parameters than a full categorical distribution over all $\boldsymbol{G} \in \mathcal{G}$.

**Definition 1** (Expressiveness of parametric families of distributions)**.** Let $\mathcal{Q}$ be a family of distributions parametrized by $\boldsymbol{\phi}$ and let $p$ be some target distribution over the same sample space $\Omega$. Then the family $\mathcal{Q}$ is $(D, \Delta)$-expressive for a distribution $p$, if there exists $q \in \mathcal{Q}$ that has a statistical divergence $D$ equal or less than $\Delta \geq 0$, i.e., $\min_{q_{\boldsymbol{\phi}} \in \mathcal{Q}} D(q_{\boldsymbol{\phi}}\|p) \leq \Delta$.

In the following, we present different models for distributions over DAGs proposed in recent works and investigate their expressiveness. For clarity and comprehensibility, we introduce these distributions by their generative model and provide the corresponding probability mass functions in Appendix A. As a summary, we visualize their generative models in Figure 3 and provide a overview of the figures, equations and the number of learnable parameters for each model in Table 1. To avoid limitations by specific functional relationships or subjective prior knowledge, we consider independence relations as a general desideratum. In particular, we begin the investigation of the expressiveness of the distributions over DAGs with the following MEC which will serve as a running example.

**Example 1.** For a parametric linear model with exogenous Gaussian noise, the true causal graph is identifiable only up to its MEC (Peters et al., 2017). Consider the simple case with three variables and the chain graph $X_1 - X_2 - X_3$ as the MEC. Its edges cannot be oriented—even in the limit of infinite data. To represent the corresponding uncertainty over the true DAG, all graphs of this MEC depicted in Figure 2, the common cause, $\boldsymbol{G}^{(1)}$, and the two causal chains, $\boldsymbol{G}^{(2)}$ and $\boldsymbol{G}^{(3)}$, should be assigned the probability $\frac{1}{3}$.

### 3.1 Independent edges

The arguably simplest probabilistic model for a distribution over a directed graph $\boldsymbol{A} \in \mathcal{A}$—not necessarily acyclic—consists of a product over independent Bernoulli probabilities, each of them modeling a possible directed edge $X_i \to X_j$:

$$\boldsymbol{A} \sim q_{\boldsymbol{\phi}}(\boldsymbol{A}) \qquad \text{with} \quad A_{ij} \sim q_{\phi_{ij}}(A_{ij}) \,, \tag{3}$$

Table 1: Overview of the different considered distributions over DAGs alongside their number of learnable parameters which depends on the number of variables $D$, particles $K$, hidden neurons $H_N$ as well as embedding and key size $H_E$ and $H_K$ (their used default values are reported in section D.2)

| Graph model | Figure | Equations | | Number of learnable parameters |
|---|---|---|---|---|
| | | Generative model | Distribution | |
| ProDAG | 3a | (4) | (22) | $D(D-1)$ |
| Binary adjacency matrix | 3b | (5) | (23) | $D(D-1)$ |
| Graph particles | 3c | (6) | (24) | $K\left(D(D-1)+1\right)$ |
| DPM-DAG | 3d | (8) | (26) | $D + D(D-1)$ |
| ARCO-DAG | 3d | (8),(9) | (26) | $H_N(D^2+1)+(H_N+1)D+D(D-1)$ |
| Mixture models | 3e | (8), (9) | (25) | $K(M+1)$ |
| GFlowNet-DAG | 3f | (10) | (29) | $2DH_E + 7(H_LD^2 + 16(H_EH_K + H_K^2)) + 4H_E^2$ |

where $\boldsymbol{\phi}$ are the parameters of the Bernoulli distributions over the edges. While self-loops can be directly ruled out by setting $\forall i \in [D] : \mathrm{A}_{ii} = 0$, the random graphs $\mathbf{A}$ drawn from this distribution can still have cycles. While rejection sampling can be used to constrain the outcome space to DAGs $\mathcal{G}$, it is highly sample-inefficient. Modeling a random weighted adjacency matrix $\mathbf{W} \in \mathcal{W} \coloneqq \mathbb{R}^{D \times D}$ instead of a binary one allows to define a generative model over a unique projection of continuous samples to $\mathcal{G}$ (Thompson et al., 2024),

$$\mathbf{W} \sim q_{\boldsymbol{\phi}}(\boldsymbol{W}) \qquad \text{with} \quad \mathrm{W}_{ij} \sim q_{\phi_{ij}}(W_{ij}), \qquad \mathbf{G} = \mathrm{Proj}_{h,\delta}(\mathbf{W}) \, , \tag{4}$$

that is depicted in Figure 3a. It is based on the idea of a continuous relaxation of the discrete optimization problem over $\mathcal{G}$ and enforces acyclicity of a single weighted adjacency matrix using a nonnegative, differentiable constraint function $h : \mathcal{W} \to \mathcal{R}_0^+$ that evaluates to zero for any acyclic graph and to some positive value quantifying the deviation, e.g., number of closed cycles (Zheng et al., 2018; Yu et al., 2019; Bello et al., 2022; Nazaret et al., 2024) otherwise. To avoid sampling only fully connected DAGs, thresholding by some parameter $\delta$ can be applied element-wise to the projected continuous adjacency matrix subsequently. Since the resulting projection $\mathrm{Proj}_{h,\delta}$ has to be computed for each sample $\mathbf{W}$, the probability of some target graph under this model cannot be evaluated directly, but estimated using Monte Carlo sampling.

A probabilistic model that allows for analytic odds to compare the probability of two graphs and avoids rejection sampling, can be derived by introducing $h$ within an exponential prefactor to the unconstrained probability $q_{\boldsymbol{\phi}}(\boldsymbol{A})$ of Eq. (3), i.e.,

$$q_{\boldsymbol{\phi},\lambda}(\boldsymbol{G}) \propto \exp(-\lambda h(\boldsymbol{G})) \, q_{\boldsymbol{\phi}}(\boldsymbol{G}) \, . \tag{5}$$

For a sufficiently high prefactor $\lambda$, the resulting distribution $q_{\boldsymbol{\phi},\lambda}(\boldsymbol{G})$ assigns only negligible probability mass to any cyclic graph. We provide the corresponding graphical model in Figure 3a. For an independent factorization over the edges of the graph, this comes at the cost of expressiveness, since the resulting model can only learn a single ordering of the nodes (Rittel & Tschiatschek, 2024). In Example 1, each chain implies a total order and the common cause a partial order, all of them being incompatible with each other. Hence, such distribution with independent edge probabilities can only concentrate its probability mass on one of the three graphs resulting in a skewed uncertainty measure.

## 3.2 Particle distribution

Lorch et al. (2021) circumvent the limitation of being constrained to a single order by modeling the graph posterior $p(\boldsymbol{G}|\mathcal{D})$ by a particle distribution consisting of a set of $K$ particles, each representing a single DAG $\boldsymbol{G}^{(k)}$. The term "particle distribution" originates from the idea of approximating a continuous density function by discrete probability masses. In the context of approximating a discrete distribution, it rather refers to a simplified model that constrains the support to fewer possible outcomes, e.g., $|\mathcal{G}| > K$ DAGs. The corresponding generative model is visualized in Figure 3c and can be expressed as

$$\mathrm{k} \sim q_{\boldsymbol{w}}(k), \quad \mathbf{G} = \boldsymbol{G}^{(\mathrm{k})} \, , \tag{6}$$

where $q_{\boldsymbol{w}}$ is a categorical distribution with $K$ possible outcomes and weights $\boldsymbol{w}$ defining the probability of each particle/DAG.

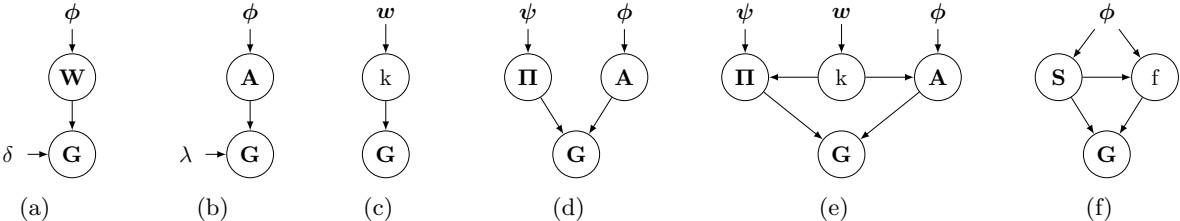

Figure 3: Generative models of the considered candidate distributions over DAGs in section 3. *(a)* Projection of a sampled continuous adjacency matrix $\mathbf{W}$, *(b)* constrained binary random matrix $\mathbf{A}$, *(c)* K graph distribution extending the former model by a categorical distribution over k matrices, *(d)* order-based model that masks $\mathbf{A}$ by some random permutation matrix $\mathbf{\Pi}$, *(e)* K order-based model, *(f)* sequence-based model that adds edges autoregressively until the finalization signal f = 1 is sampled or the DAG is complete.

### 3.3 Mixture models

More expressive distributions over DAGs can be defined by replacing the $K$ particles from section 3.2, each corresponding to a specific graph, by $K$ distributions over DAGs, i.e.,

$$\text{k} \sim q_{\boldsymbol{w}}(k)\,, \quad \mathbf{G} \sim q_{\boldsymbol{\phi}^{(\text{k})}}(\boldsymbol{G})\,. \tag{7}$$

Consequently, we no longer refer to the resulting distribution as a particle distribution but as a probabilistic mixture model.

In principle, Eq. (5) could be used as the base model allowing to model $K$ different total orders of the nodes instead of a single one[1]. In the running Example 1 the stated MEC consists only of 3 graphs, but each of them is admissible under a different total order. This illustrates that here at least $K = 3$ is required. In case the distribution concentrates its probability mass on graphs that are admissible for a low number of orders, e.g., sparse graphs with multiple components, such a model may suffice. Although smaller than the number of possible DAGs, the number of total orders and permutations $D!$ is still super-exponential. Hence, for a higher number of variables only a very small fraction of all permutations can be represented by the resulting mixture model. Evidently, having a single probabilistic model for each total order constitutes a poor trade-off between expressiveness and efficiency due to the increased number of trainable model parameters. To address this limitation, we consider probabilistic models that explicitly allow for different orderings in the following without enumerating all of them.

### 3.4 Probabilistic models over orders

Several works apply the idea of an order-based search (Friedman & Koller, 2003; Teyssier & Koller, 2005) in a probabilistic generative model (Cundy et al., 2021; Charpentier et al., 2022; Wang et al., 2022; Rittel & Tschiatschek, 2023; Annadani et al., 2023). Their shared underlying idea is to learn a total order of the variables that imposes an acyclicity constraint on the (sampled) adjacency matrix $\mathbf{A}$. Following the description of DPM-DAG (Rittel & Tschiatschek, 2023), we outline its generative model and provide the corresponding graphical model in Figure 3d. The generative process starts by drawing a total order that defines a permutation $\mathbf{\Pi}$ of the variables. A permuted upper-triangular matrix of ones $M$ then defines a random acyclicity matrix $\mathbf{M^{(\Pi)}}$ that is used to mask a sample of an unconstrained adjacency matrix $\mathbf{A}$ as modeled in Eq. (5), i.e.,

$$\mathbf{\Pi} \sim q_{\psi}(\mathbf{\Pi})\,, \quad \mathbf{A} \sim q_{\phi}(\boldsymbol{A})\,, \quad \mathbf{G} = \underbrace{\left(\mathbf{\Pi}^T M \mathbf{\Pi}\right)}_{=:\,\mathbf{M^{(\Pi)}}} \circ \mathbf{A}\,. \tag{8}$$

This masking by $\mathbf{M^{(\Pi)}}$ can be interpreted as a probabilistic projection of $\mathbf{A}$ to $\mathbf{G}$. In contrast to Eq. (4), $q_{\boldsymbol{\psi},\boldsymbol{\phi}}(\boldsymbol{G})$ can be efficiently evaluated as $\mathbf{\Pi}$ and $\mathbf{A}$ are independent random matrices with analytic probability

---

[1]This correspond to the model by Lorch et al. (2021) without collapsing the $K$ distributions to $K$ single graphs.

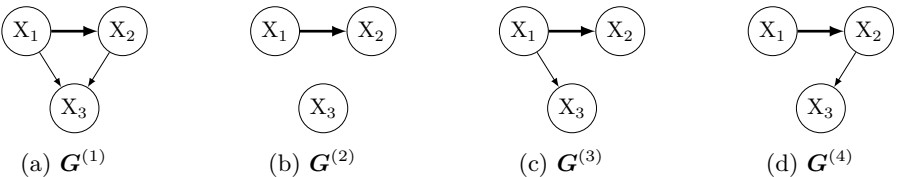

Figure 4: Graphs of Example 2 that all share the highlighted edge between $X_1$ and $X_2$.

mass functions. For details we refer to section A.6. The distribution over the permutation can be modeled by the *Plackett-Luce* (PL) distribution (Plackett, 1975), but comes with a limitation for causal discovery (Rittel & Tschiatschek, 2023; Toth et al., 2024) that can be demonstrated with Example 1. Both chains, $\boldsymbol{G}^{(2)}$ and $\boldsymbol{G}^{(3)}$, imply a total order of the three variables, but reversed ones. In the PL model, their permutations can only receive the same probability mass in the case of uniform weights for all three variables, yet the probability for the two causal chains is then upper-bounded by $\frac{1}{6}$ and cannot take the value of $\frac{1}{3}$. Some remaining probability mass is then concentrated on other graphs that do not belong to the MEC. This motivated the ARCO-DAG model (Toth et al., 2024), an autoregressive model over causal orders that computes the weights for the categorical sampling without replacement at each sampling stage conditionally on the previous drawn sequence:

$$q_{\boldsymbol{\psi}}(\pi) = \prod_{d=1}^{D} q_{\boldsymbol{\psi}^{(d)}}\big(\pi(d)\big) \qquad \text{with} \quad \boldsymbol{\psi}^{(d)} = f_{\boldsymbol{\psi}}\left(\{\pi(i)\}_{i=1}^{d}\right) \ . \tag{9}$$

The function $f_{\boldsymbol{\psi}} : \mathcal{R}^{D \times D} \mapsto [D]$ takes a permutation matrix where some rows are still zeros as input and predicts the weights for sampling the next variable in the order. For their experiments, the authors of ARCO applied a multilayer perceptron with a single hidden layer with $H > D$ neurons. To avoid marginalizing over the set of possible parents under a sampled permutation as in Toth et al. (2024), the same distribution over an unconstrained graph may be used yielding Eq. (8) but with an autoregressive distribution over the permutation. In the case of Example 1 where the probability mass should be split equally on all graphs of the MEC $X_1 - X_2 - X_3$, this is consistent with setting $P_{\mathbf{A}_{13}}(1) = P_{\mathbf{A}_{31}}(1) = 0$ and $P_{\mathbf{A}_{ij}}(1) = 1$ otherwise. To further increase the expressiveness of DPM- and ARCO-DAG, mixtures of both can be considered. The resulting graphical model is depicted in Figure 3e.

### 3.5 Fully autoregressive model

In all covered candidate models for distributions over DAGs, the unmasked edges are still sampled from independent Bernoulli distributions as stated in Eq. (5) or relaxations of them, e.g., Gumbel-Softmax distribution Maddison et al. (2017); Jang et al. (2017). Besides acyclicity, the edges of the causal graph drawn from these distributions are not coupled—limiting the models' expressiveness.

**Example 2** (Coupled edges)**.** Consider the four DAGs depicted in Figure 4 with $\boldsymbol{G}^{(1)}$ being the true causal graph. Assume the functional relationship between $X_1$ and $X_2$ is identifiable and, hence, the cumulative probability of all graphs that contain this edge should be very high in the posterior. If the structural equation for variable $X_3$, $f_{X_3}(X_1, X_2, \epsilon_3)$, depends only on both variables $X_1$ and $X_2$, but not a single one alone, e.g., $f_3 = \frac{X_1 X_2}{1 + X_1^2 + X_2^2} + \epsilon_3$ , then the corresponding edges are coupled. Consequently, the likelihoods of $\boldsymbol{G}^{(2)}$, $\boldsymbol{G}^{(3)}$ and $\boldsymbol{G}^{(4)}$ are almost the same, but are all lower than the one of the true graph $\boldsymbol{G}^{(1)}$. Favoring sparser graphs, e.g., by the prior, can then lead to a posterior that concentrates its probability mass on $\boldsymbol{G}^{(1)}$ and $\boldsymbol{G}^{(2)}$ that misses $X_1$ or $X_3$ as a single cause of $X_2$. The graphs $\boldsymbol{G}^{(3)}$ and $\boldsymbol{G}^{(4)}$ with only $X_1$ or $X_2$ as causes of $X_3$ should be assigned small probability in this setting.

To enable arbitrary dependencies between edges, Deleu et al. (2022) propose to construct DAGs sequentially by adding edges to DAGs with fewer edges, starting with the empty graph $\boldsymbol{G}_0$. Their generative model, to which we refer to as GFlowNet-DAG, is motivated by the following rationale: Denote $E := |\boldsymbol{G}| \in \mathbb{N}_0$ as the number of edges of a DAG. Then a sampled (possibly empty) sequence of distinct edges $\boldsymbol{S} = (S_1, \ldots, S_E) \in \mathcal{S}$ uniquely defines a DAG $\boldsymbol{G}$ over a mapping $g : \mathcal{S} \mapsto \mathcal{G}$, although $E!$ different sequences can yield the same

graph $G$. The underlying sampling of the GFlowNet-DAG model is based on a transformer architecture $t_{\phi} : \mathcal{G} \mapsto \mathcal{R}^{D \times D}$ that autoregressively computes the parameters $\varphi$ defining the probability for discontinuation of the generative process as well as the probabilities for the potential new edges, $\varphi := t_{\phi}(S_{:i-1})$. Each construction step $i$ begins with sampling of a binary signal $f_i$ that indicates whether the graph prototype $G_{i-1}$ is final, $f = 1$, and the generative process stops. Otherwise, $f = 0$, a new edge $S_i$ is sampled and added to the current graph $G_i$:

$$f_i \sim q_{\phi}(f_i | \mathbf{S}_{:i-1}), \quad \mathbf{S}_i \sim q_{\phi}(\mathbf{S}_i | \mathbf{S}_{:i-1}), \quad \mathbf{G}_i = g(\mathbf{S}_{:i}) . \tag{10}$$

During the construction of the DAG, acyclicity is enforced by masking any edges that would create a cycle, i.e., the transitive closure of the adjacency matrix of the graph that decodes ancestral relations. After the addition of a new edge, the ancestral mask is updated and the iterative sampling continues until the stop signal $f_i = 1$ is sampled or the DAG is fully connected. The corresponding graphical model is shown in Figure 3f. The discontinuation signal $\boldsymbol{f}_i$ is a necessary requirement, since it guarantees that each DAG can be assigned an arbitrary probability mass provided that the function $t_{\phi}$ has the capacity to model the transition probabilities between different states exactly.

## 4 Evaluation

In contrast to Bayesian causal discovery algorithms where an observed data set is the basis for unsupervised learning of the causal graph, in this work we focus on the comparison of different models for distributions over DAGs w.r.t. their expressiveness for known target distributions as outlined in Def. 1. This supervised setting allows us to shield off several sources of error, including any bias due to the choice of functional relationships, prior distributions over the data, or the size of the observed training data.

To demonstrate the limitations of the candidate models for distribution over DAGs from section 3, we fit each candidate model $q_{\boldsymbol{G}}$ to a specified target distribution $p_{\boldsymbol{G}}$. The target distribution is either derived from the MEC in Example 1, by the coupling of edges from Example 2, or a synthetically generated distribution that arises from concentrating the probability mass around a target graph based on the structural Hamming distance (SHD).

For training of the parameters $\boldsymbol{\phi}$ with gradient descent, we take the forward Kullback-Leibler (KL) divergence between the target distribution $p_{\mathbf{G}}$ and the candidate distribution $q_{\mathbf{G}}$ as the loss function and approximate it using samples from the target distribution. Due to the limited support of particle distributions, we evaluate the fitted candidate distribution with the reverse KL divergence $D_{\mathrm{KL}}(q_{\boldsymbol{G}} \| p_{\boldsymbol{G}})$ and the total variation distance $D_{\mathrm{TV}}(q_{\boldsymbol{G}} \| p_{\boldsymbol{G}})$.

In contrast to its backward formulation, the forward KL divergence estimates the log-likelihood ratio using samples from the target distribution. During supervised training with gradient descent, this ensures finding the region of the support where the probability mass is concentrated. To illustrate this important detail consider the GFlowNet model where the generation of a DAG starts with the empty graph. If the probability mass is concentrated around graphs with a moderate number of edges, almost no signal for learning is provided for graphs with only one or two of its edges.

### 4.1 Particle distributions

For particle distributions, we can compute their optimal statistical divergences to general discrete distributions over a finite set of outcomes.

**Lemma 1** (Minimal statistical distances for particle distributions). *The reverse Kullback-Leibler divergence, the total variation, the Hellinger distance, and the Bhattacharyya distance between a discrete target distribution $p_{\mathbf{G}}$ and a particle representation $q_{\mathbf{G}}$, as defined in Eq. (6), are minimized, if the particles of $q_{\mathbf{G}}$ represent the graphs with the highest probability mass in the target distribution $p_{\mathbf{G}}$ and have their normalized*

*probability mass:*

$$q_{\mathbf{G}}(\mathbf{G}) = \sum_{k=1}^{K} \left[ \boldsymbol{G} = \boldsymbol{G}^{(k)} \right] q_{\mathrm{k}}(k) \,, \tag{11}$$

$$with \quad \forall k \in \{1, \ldots, K\}: \qquad \boldsymbol{G}^{(k)} \coloneqq \underset{\boldsymbol{G} \in \mathcal{G} \setminus \{\boldsymbol{G}^{(i)}\}_{i<k}}{\arg\max} \, p(\boldsymbol{G}) \,, \qquad q(\boldsymbol{G}^{(k)}) \coloneqq \frac{p(\boldsymbol{G}^{(k)})}{\sum_{j=1}^{K} p(\boldsymbol{G}^{(j)})} \,. \tag{12}$$

The minimal reverse KL divergence and total variation distance are given by:

$$\min_{q \in \mathcal{Q}} D_{\mathrm{KL}}(q_{\mathbf{G}} \| p_{\mathbf{G}}) = -\log \sum_{k=1}^{K} p(\boldsymbol{G}^{(k)}) \,, \tag{13}$$

$$\min_{q \in \mathcal{Q}} D_{\mathrm{TV}}(q_{\mathbf{G}} \| p_{\mathbf{G}}) = 1 - \sum_{k=1}^{K} p(\boldsymbol{G}^{(k)}) \,. \tag{14}$$

We present the full derivation of all minima and the proof of Lemma 1 in Appendix B.

## 4.2 Semi-implicit generative models

A latent generative model for $\mathbf{G}$ defines a sampling procedure that starts with sampling some auxiliary random objects $\mathbf{Y} \sim q_{\boldsymbol{\phi}}$. A function $g$ then maps the drawn random samples $\mathbf{Y}$ to a DAG, i.e., $\mathbf{G} = g(\mathbf{Y})$, and induces a joint distribution for $q(\mathbf{G}, \mathbf{Y})$. For DAGs, $\mathbf{Y}$ can for instance be a continuous adjacency matrix as in Eq. (4), a combination of a permutation and an unconstrained binary adjacency matrix as in Eq. (8) or a sequence of edges $\mathbf{S}$ as in Eq. (10). For such an implicit distribution, the probability of some given graph $\boldsymbol{G}^*$, e.g., sampled from the target distribution, cannot be easily evaluated since $g$ is in general not injective. The joint distribution has to be marginalized either by analytic integration/explicit summation over $\mathbf{Y}$ or approximated by samples of $\mathbf{Y}$, i.e.,

$$q_{\boldsymbol{\phi}}(\boldsymbol{G}^*) = \int \left[ \boldsymbol{G}^* = g(\boldsymbol{Y}) \right] q_{\boldsymbol{\phi}}(g(\boldsymbol{Y}), \boldsymbol{Y}) \, \mathrm{d}\boldsymbol{Y} \,. \tag{15}$$

Direct sampling of $\mathbf{G}$ over $\mathbf{Y}$ has relatively high variance for graphs with low probability. Moreover, optimization of the parameters of a candidate distribution by gradient descent with samples from the target distribution $p$ is costly, since the update requires computing the probability for all drawn samples from the candidate distribution $q$. As a remedy, we propose to use importance sampling to efficiently estimate the marginal probability of $\boldsymbol{G}^*$ using discrete samples $\mathbf{Y}^*$ from the proposal distribution $q_{\boldsymbol{\phi}}^*$ that only generate the target graph $\boldsymbol{G}^*$:

$$q_{\boldsymbol{\phi}}(\boldsymbol{G}^*) = \sum_{\boldsymbol{Y}^* : \, g(\boldsymbol{Y}^*) = \boldsymbol{G}^*} q_{\boldsymbol{\phi}}(\boldsymbol{G}^*, \boldsymbol{Y}^*) = \mathbb{E}_{(\boldsymbol{G}^*, \mathbf{Y}^*) \sim q_{\boldsymbol{\phi}}^*} \left[ \frac{q_{\boldsymbol{\phi}}(\boldsymbol{G}^*, \mathbf{Y}^*)}{q_{\boldsymbol{\phi}}^*(\boldsymbol{G}^*, \mathbf{Y}^*)} \right] \,. \tag{16}$$

**Lemma 2** (Sample-efficiency of the IS estimator)**.** *Assume that $q(\boldsymbol{G}, \boldsymbol{Y})$ is defined by a generative model which produces a random DAG $\mathbf{G}$ as a function $g$ for some sampled discrete variables $\mathbf{Y}$ and partitions the space of DAGs. The importance-sample estimator $\hat{q}_{IS}(\boldsymbol{G}^*)$ derived from Eq. (16) allows directly drawing samples of $\boldsymbol{G}^*$ and has a smaller variance than standard Monte Carlo sampling, i.e., its effective sample size is greater than $N$, the number of drawn samples.*

Note that continuous adjacency matrices do not directly partition the discrete space of DAGs allowing to analytically evaluate the probability of a target graph. Hence, we focus our investigation on order- and sequenced-based models where $\mathbf{Y}$ is discrete and the optimal proposal distribution $q_{\boldsymbol{\phi}}^*$ can be derived directly from the distribution $q_{\boldsymbol{\phi}}$ by constraining $\mathbf{Y}$ to $\boldsymbol{Y}^*$, e.g., evaluating only admissible permutations and unmasked parts of an adjacency matrix. Due to the parameter sharing of $\boldsymbol{\phi}$, the proposal distribution is not constant over training and is derived from the candidate (target) distribution at every optimization step.
We provide the proof of Lemma 2 and details on the implementation in Appendix C. While we present the proposed importance sampling to evaluate the probability of a full graphs, its application to directed paths or subgraphs is straight-forward.

# 5  Experiments

In our experiments, we compare the following models from section 3 and Table 1 in a supervised setting:

- *K graph particle distributions.* Distributions over a fixed number of DAGs.

- *Reinforced probabilistically masked DAGs (RPM-DAG).* A discretized version of DPM-DAG (Rittel & Tschiatschek, 2023) that we refer to as RPM-DAG, since the score-function gradient estimator is applied instead of pathwise Gumbel-SoftSort gradient estimation,

- *ARCO-DAG.* An autoregressive model over the causal order (Toth et al., 2024),

- *Mixture models of RPM-DAGs/ARCO-DAGs.* A categorical distribution over $K$ order-based models as depicted in Figure 3e (with $K = 1$ equals the base model).

- *GFlowNet-DAG.* A GFlowNet that generates DAGs by the sequential addition of edges starting from the empty graph (Deleu et al., 2022).

In the supervised setting, we minimize the forward KL divergence between the target distribution and the model distribution using the Adam Optimizer with decoupled weight decay (Loshchilov & Hutter, 2019) over 1000 optimization steps. Further details and the used hyperparameters for each model are provided in section D.2.

## 5.1  Markov equivalent graphs

The first experiment consists of the MEC represented by the chain graph $A - B - C$ motivated in Example 1. The target distribution assigns all three graphs of the equivalence class the same probability, i.e., $\frac{1}{3}$.

In Table 2a we report the mean of the reverse KL divergence and the total variation distance together with their standard deviations error over 20 independent training and evaluation runs for the RPM-DAG, ARCO-DAG, and GFlowNet-DAG model. For the particle distributions we compute the optimal empirical values according to Eq. (13), (14), and (12).

The results align with the theoretical analysis that RPM-DAG fails to distribute the probability equally among all graphs of the MEC. The observed preference for the common cause $\boldsymbol{G}^{(1)}$ is a model bias arising from the fact that $\boldsymbol{G}^{(1)}$ induces only a partial order that is compatible with two total ones.

For $K = 3$, a particle distribution can perfectly match the target distribution. For fewer particles, this matching is not possible and the particle distribution is outperformed by ARCO-DAG and GFlowNet-DAG. However, our experimental result for the mixture of 3 RPM-DAG models indicates that these optimal theoretical values are not reached during model training. With an increasing number of components in the mixture models, the statistical divergence only decreases slowly, while their standard deviations increase initially, before they decrease again. The standard errors underline the validity of the reported mean values, since they are by a factor of $\sqrt{20}$ smaller than the reported standard deviations. The high variance is due to some outliers, illustrating that there is a regime in which the training is not stable.

In contrast, the results for ARCO-DAG demonstrate that it can overcome the limitations of RPM-DAG by using a simple autoregressive model for the permutation weights. In comparison, to the GFlowNet-DAG model, it yields a considerably lower total variation distance, is more stable, and is much faster in training and evaluation which can be attributed partly to the reduced number of model parameters (see Table 1).

Note that the structure consisting of only three variables also appears in graphs with more variables and comes w.l.o.g.. In the case of a graph with $C$ unconnected components, each consisting of a graph of the simple MEC, its MEC contains $3^C$ graphs. Under assumed ideal conditions and a supervised setting, a particle method picks $K$ of these graphs and assigns $\frac{1}{K}$ as probability instead of $3^{-C}$. Eq. (13) and (14) then imply high statistical divergences that are not competitive with parametrized distributions over orders, e.g., RPM-DAG, that scale well for unconnected components.

Table 2: Reverse KL divergence $D_{\mathrm{KL}}$, total variation distance $D_{\mathrm{TV}}$, and graph probabilities $q(\boldsymbol{G})$ for different candidate models $q_{\mathbf{G}}$ and target distribution $p_{\mathbf{G}}$. Empirical means and standard deviations are computed from 20 independent runs, except for the particle representations for which their analytical optimal values are computed using Eq. (13) and (14).

(a) Example 1: MEC class $\mathrm{X_1 - X_2 - X_3}$ .

| $q_{\mathbf{G}}$ | $D_{\mathrm{KL}}(q_{\mathbf{G}}\|p_{\mathbf{G}})\downarrow$ | $D_{\mathrm{TV}}(q_{\mathbf{G}}\|p_{\mathbf{G}})\downarrow$ | $q(\boldsymbol{G}_1)$ | $q(\boldsymbol{G}_2)$ | $q(\boldsymbol{G}_3)$ |
|---|---|---|---|---|---|
| $p_{\mathbf{G}}$ | 0 | 0 | $0.\overline{3}$ | $0.\overline{3}$ | $0.\overline{3}$ |
| 1 graph particle | 1.099 | $0.\overline{6}$ | 1. | 0 | 0 |
| 2 graph particles | 0.406 | 0.3 | 0.5 | 0.5 | 0 |
| 3 graph particles | 0 | 0. | $0.\overline{3}$ | $0.\overline{3}$ | $0.\overline{3}$ |
| RPM-DAG | $0.32673 \pm 0.00004$ | $0.34377 \pm 0.00003$ | $0.53318 \pm 0.00001$ | $0.16145 \pm 0.00002$ | $0.16145 \pm 0.00002$ |
| 5-RPM-DAG | $0.05 \quad \pm 0.12$ | $0.05 \quad \pm 0.11$ | $0.36 \quad \pm 0.07$ | $0.31 \quad \pm 0.06$ | $0.07 \quad \pm 0.06$ |
| 10-RPM-DAG | $0.0013 \pm 0.0009$ | $0.003 \pm 0.008$ | $0.335 \pm 0.007$ | $0.333 \pm 0.001$ | $0.331 \pm 0.009$ |
| ARCO-DAG | $0.00370 \pm 0.00005$ | $0.0038 \pm 0.0003$ | $0.3322 \pm 0.0007$ | $0.3320 \pm 0.0007$ | $0.3321 \pm 0.0006$ |
| 5-ARCO-DAG | $0.0021 \pm 0.0006$ | $0.003 \pm 0.004$ | $0.333 \pm 0.004$ | $0.332 \pm 0.005$ | $0.333 \pm 0.002$ |
| 10-ARCO-DAG | $0.0015 \pm 0.0006$ | $0.0220 \pm 0.0021$ | $0.333 \pm 0.002$ | $0.3327 \pm 0.0017$ | $0.3324 \pm 0.0012$ |
| GFlowNet-DAG | $0.004 \quad \pm 0.015$ | $0.02 \quad \pm 0.04$ | $0.328 \pm 0.026$ | $0.34 \quad \pm 0.04$ | $0.328 \pm 0.023$ |

(b) Example 2: Coupled edges $\mathrm{X_1 \rightarrow X_3 \leftarrow X_2}$ and identifiable causal effect $\mathrm{X_1 \rightarrow X_2}$ .

| $q_{\mathbf{G}}$ | $D_{\mathrm{KL}}(q_{\mathbf{G}}\|p_{\mathbf{G}})\downarrow$ | $D_{\mathrm{TV}}(q_{\mathbf{G}}\|p_{\mathbf{G}})\downarrow$ | $q(\boldsymbol{G}_1)$ | $q(\boldsymbol{G}_2)$ | $q(\boldsymbol{G}_3) + q(\boldsymbol{G}_4)$ |
|---|---|---|---|---|---|
| $p_{\mathbf{G}}$ | 0 | 0 | 0.3 | 0.6 | 0.1 |
| 1 graph particle | 0.5108 | 0.4 | 0 | 1 | 0 |
| 2 graph particles | 0.1054 | 0.1 | $0.\overline{3}$ | $0.\overline{6}$ | 0 |
| 3 graph particles | 0.0513 | 0.05 | 0.3158 | 0.6316 | 0.0526 |
| 4 graph particles | 0 | 0 | 0.3 | 0.6 | 0.1 |
| RPM-DAG | $0.333 \quad \pm 0.009$ | $0.36 \quad \pm 0.01$ | $0.126 \quad \pm 0.012$ | $0.414 \quad \pm 0.023$ | $0.44 \quad \pm 0.02$ |
| 2-RPM-DAG | $0.0024 \pm 0.0008$ | $0.0046 \pm 0.0047$ | $0.2986 \pm 0.0006$ | $0.5970 \pm 0.0046$ | $0.1004 \pm 0.0011$ |
| ARCO-DAG | $0.328846 \pm 0.000002$ | $0.355704 \pm 0.000012$ | $0.122402 \pm 0.000015$ | $0.42189 \pm 0.00003$ | $0.45447 \pm 0.00002$ |
| 2-ARCO-DAG | $0.00126 \pm 0.00005$ | $0.0021 \pm 0.0017$ | $0.2999 \pm 0.0020$ | $0.5993 \pm 0.0020$ | $0.1004 \pm 0.0002$ |
| GFlowNet-DAG | $0.0001 \pm 0.0003$ | $0.004 \quad \pm 0.004$ | $0.30 \quad \pm 0.05$ | $0.599 \pm 0.004$ | $0.099 \pm 0.004$ |

## 5.2 Coupled edges

Dependent edges in the posterior as in Example 2 motivate the second experiment in which we consider the corresponding graphs of Figure 4 with the following probabilities. The target distribution concentrates 60% of the probability mass on the graph $\boldsymbol{G}^{(2)}$ that misses $\mathrm{X_1}$ and $\mathrm{X_2}$ as parents of $\mathrm{X_3}$, further 30% are assigned to the true causal graph $\boldsymbol{G}^{(1)}$. To represent a coupling of the two causes, the graphs $\boldsymbol{G}^{(3)}$ and $\boldsymbol{G}^{(4)}$ that miss either of these parents get only a probability of 0.05 each.

In Table 2b the corresponding metrics for 20 runs are listed, following the same reporting as in section 5.1. Both RPM-DAG and ARCO-DAG, assign the graphs $\boldsymbol{G}^{(3)}$ and $\boldsymbol{G}^{(4)}$ too high probability as they fail to account for the coupling of the edges $\mathrm{X_1 \rightarrow X_3}$ and $\mathrm{X_2 \rightarrow X_3}$. Due to its autoregressive model over edges, GFlowNet-DAG can approximate the target posterior with very high accuracy in terms of individual graph probabilities as well as the two evaluated statistical divergences. By design, a particle distribution is not constrained by dependent edges. Since 90% of the probability is concentrated on two graphs, a reasonable approximation can be obtained by modeling only two graphs in this particular example. However, our results show that both probabilistic mixture models, with two components of RPM- or ARCO-DAG models each, outperform a distribution with 3 graphs particles.

Table 3: Target distribution concentrated around the MAP graph $\boldsymbol{G}_\dagger$. Graphs with the same SHD, $\boldsymbol{G} \in \mathcal{G}_{\mathrm{SHD}}$, are assigned the same probability.

| SHD | $p(\mathcal{G}_{\mathrm{SHD}})$ | $|\mathcal{G}_{\mathrm{SHD}}|$ | $p(\boldsymbol{G} \in \mathcal{G}_{\mathrm{SHD}})$ |
|---|---|---|---|
| 0 | 0.15 | 1 | 0.15 |
| 1 | 0.425 | 8 | 0.05313 |
| 2 | 0.225 | 28 | 0.00804 |
| 3 | 0.1 | 61 | 0.00164 |
| 4 | 0.05 | 94 | 0.00053 |
| 5 | 0.01581 | 111 | 0.00014 |
| 6 | 0.01439 | 101 | 0.00014 |
| 7 | 0.01068 | 75 | 0.00014 |
| 8 | 0.00627 | 44 | 0.00014 |
| 9 | 0.00242 | 17 | 0.00014 |
| 10 | 0.00043 | 3 | 0.00014 |

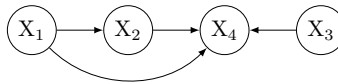

Figure 5: Assumed true graph $\boldsymbol{G}_\dagger$ representing the MAP graph in the posterior modeled in Table 3.

Table 4: Statistical divergences for different candidate models for the synthetic posterior distribution in Table 3 averaged over 20 runs.

| $q_{\mathbf{G}}$ | $D_{\mathrm{KL}}(q_{\mathbf{G}}\|p_{\mathbf{G}}) \downarrow$ | $D_{\mathrm{TV}}(q_{\mathbf{G}}\|p_{\mathbf{G}}) \downarrow$ |
|---|---|---|
| 1 graph particle | 1.8971 | 0.8500 |
| 10 graph particles | 0.5395 | 0.4170 |
| 25 graph particles | 0.3516 | 0.2964 |
| 50 graph particles | 0.1969 | 0.1787 |
| 100 graph particles | 0.1042 | 0.0989 |
| 150 graph particles | 0.0751 | 0.0723 |
| 250 graph particles | 0.0426 | 0.0417 |
| RPM-DAG | $0.133 \pm 0.008$ | $0.166 \pm 0.007$ |
| 2-RPM-DAG | $0.107 \pm 0.009$ | $0.149 \pm 0.009$ |
| 3-RPM-DAG | $0.070 \pm 0.007$ | $0.133 \pm 0.006$ |
| 4-RPM-DAG | $0.061 \pm 0.006$ | $0.123 \pm 0.006$ |
| 5-RPM-DAG | $0.057 \pm 0.005$ | $0.120 \pm 0.005$ |
| 10-RPM-DAG | $0.041 \pm 0.005$ | $0.09 \pm 0.01$ |
| 20-RPM-DAG | $0.033 \pm 0.004$ | $0.072 \pm 0.006$ |
| ARCO-DAG | $0.087 \pm 0.009$ | $0.146 \pm 0.007$ |
| 2-ARCO-DAG | $0.079 \pm 0.010$ | $0.137 \pm 0.008$ |
| 3-ARCO-DAG | $0.076 \pm 0.010$ | $0.135 \pm 0.008$ |
| 4-ARCO-DAG | $0.071 \pm 0.009$ | $0.130 \pm 0.007$ |
| 5-ARCO-DAG | $0.07 \pm 0.01$ | $0.130 \pm 0.007$ |
| 10-ARCO-DAG | $0.058 \pm 0.007$ | $0.114 \pm 0.007$ |
| 20-ARCO-DAG | $0.043 \pm 0.006$ | $0.09 \pm 0.10$ |
| GFlowNet-DAG | $0.25 \pm 0.17$ | $0.25 \pm 0.08$ |

## 5.3 Concentration of posterior mass

For an identifiable FCM and a high number of samples, the posterior should be concentrated on graphs that show a high likelihood. In the idealized setting of perfect regression of a child on its parents, the likelihood can be expected to peak for the true underlying graph $\boldsymbol{G}_\dagger$ provided that parameter uncertainty or regularization prevents superfluous edges that are not in $\boldsymbol{G}_\dagger$. This implies that high-scoring graphs are 'similar' to $\boldsymbol{G}_\dagger$ implied by the FCM.

In the absence of an analytic posterior that motivates such similarity, we generate a synthetic target distribution around the assumed *maximimum-a-posteriori* (MAP) graph $\boldsymbol{G}_\dagger$ depicted in Figure 5 that has positive support for all 543 possible DAGs with 4 nodes. We start by assigning probability masses to the groups of graphs that have a *structural Hamming distance* (SHD) up to 4, i.e., up to 4 different entries in the adjacency matrix, and split the remaining probability mass equally among all graphs with higher SHD. For the five groups with SHD up to 4, we distribute their cumulated probability masses equally among all graphs within the respective group. The target distribution is summarized in Table 3 which lists the probability masses for the groups of graphs with the same SHD alongside the individual ones.

We report in Table 4 the mean values and standard deviations for the statistical divergences over 20 independent runs for all candidate models, except the particle representations for which the optimal values are computed analytically using Eq. (13) and (14). The results highlight the expressivity of the RPM-DAG and its extension ARCO-DAG that both outperform a particle distribution consisting of the 50 graphs with the highest probability in the target distribution that account for a cumulative probability of 86.3%. A single ARCO-DAG yields lower statistical divergences than a mixture of 2 RPM-DAG models in this setting. When comparing mixture models with $K \geq 3$, the RPM-DAG mixtures show better (mean) results than ARCO-DAG. Although we trained in an idealized very low-dimensional setting with samples drawn from the target distribution, the good performance of the GFlowNet-DAG model from section 5.1 and 5.2 cannot be observed in our considered setting. Its statistical divergences are much higher than for a single RPM

or ARCO-DAG model. We conjecture that the gradients from a variety of different graphs do not yield a sufficiently stable training signal that is necessary to tune the transformer network of GFlowNet-DAG.

In addition to the results for the expressiveness of the candidate distributions, we report the training times of the candidate models and considered mixtures of them in section D.3. Notably, training the GFlowNet-DAG model takes 250 times longer than training the RPM- and ARCO-DAG.

## 6 Conclusion

Bayesian causal discovery promises uncertainty quantification in the prediction of the causal graph underlying the observed data. We reviewed several candidate models for distributions over DAGs and investigated limitations of their expressivity. To minimize confounding factors such as the influence of functional relationships or the size of the training set, we considered an idealized supervised setting that can also be used to specify prior distributions. We showed in our theoretical analysis and experimental results that all considered candidate models, except for the autoregressive GFlowNet-DAG model, are theoretically not sufficiently expressive to match simple target distributions with coupled edges. While GFlowNet-DAG is theoretically sufficiently expressive, we cannot confirm its expressiveness in a low-dimensional experiment where graph samples are drawn from the target distribution with support over all possible DAGs. Since causal structure learning is typically an unsupervised problem, this poses a major limitation of this model.

Coupled edges, due to interaction effects, pose a general challenge in causal structure learning and affect most algorithms—not only Bayesian approaches. Our results suggest that a mixture of a small number of simple probabilistic models such as the RPM-DAG model may approximate distributions with coupled edges sufficiently well in practical applications and outperform particle distributions with a moderate number of modeled graphs. Its extension, ARCO-DAG, is more expressive w.r.t. probabilistic causal orders and shows competitive performance to GFlowNet-DAG, while having a substantially lower number of parameters, and is more stable and by a magnitude faster in training. When comparing RPM- and ARCO-DAG mixture models, the RPM-DAG mixtures outperform the ARCO-DAG mixtures for 10 mixture components consistently in all experiments. Therefore, we conjecture that a mixture of RPM-DAG models is particularly suited to scale Bayesian causal discovery with higher numbers of variables.

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

# A Probability mass functions over DAGs

For conciseness, we presented in section 3 only the generative models for DAGs that induce a probability distribution. In the following, we provide their analytic probability mass functions.

## A.1 Unconstrained, binary adjacency matrix with independent edges

Sampling an off-diagonal adjacency matrix $\mathbf{A}$ with independent edges as in Eq. (3), does not guarantee that the sampled graph is a DAG. Rejecting all acyclic graphs implicitly defines a distribution whose support is limited to $\mathcal{G}$. Even for moderate values of individual edge probabilities $q_{\mathbf{A}_{ij}}$, rejection sampling can be highly sample-inefficient, in particular for graphs with many nodes, i.e., higher values of $D$, that allow for more cycles. To evaluate any given DAG $\boldsymbol{G}$, renormalization of the probability of $q_{\mathbf{A}}(\boldsymbol{G})$ by the constant $Z$ is required which can be approximated by the relative frequency that a sampled graph is a DAG:

$$q_{\mathbf{A}}(\boldsymbol{A}) = \prod_{i \neq j}^{D} q_{\mathrm{A}_{ij}}(A_{ij}) \,, \tag{17}$$

$$q_{\mathbf{G}}(\boldsymbol{G}) = \frac{1}{Z} q_{\mathbf{A}}(\boldsymbol{G}) \qquad \text{with} \ \ Z := \sum_{\boldsymbol{A} \in \mathcal{A}} \left[\boldsymbol{A} \in \mathcal{G}\right] q_{\mathbf{A}}(\boldsymbol{A}) = \mathbb{E}_{\mathbf{A} \sim q_{\mathbf{A}}} \left[\boldsymbol{A} \in \mathcal{G}\right] \,. \tag{18}$$

However, using the unnormalized distributions $q_{\mathbf{A}}$ allows to evaluate the odds of two DAGs without any sampling or computation of normalization constant $Z$.

## A.2 Projection of samples from an unconstrained, continuous adjacency matrix

To avoid costly rejections of cyclic graphs in the generative process, a continuous adjacency matrix $\mathbf{W}$ drawn from the generative model defined in Eq. (4) can be projected on the space of acyclic, weighted adjacency matrices $\mathcal{V} := \{\boldsymbol{W} \in \mathcal{W} \mid h(\boldsymbol{W}) = 0\}$:

$$\boldsymbol{V} = \mathrm{Proj}_h(\boldsymbol{W}) := \operatorname*{arg\,min}_{\boldsymbol{V} \in \mathcal{V}} \frac{1}{2} \|\boldsymbol{W} - \boldsymbol{V}\|_{\mathrm{F}}^2 \,. \tag{19}$$

As the corresponding weighted DAG $\boldsymbol{V} \in \mathcal{V}$ is still fully connected, L1 regularization can be applied to model sparser graphs. Finally, to obtain a binary DAG $\boldsymbol{G} \in \mathcal{G}$ from $\boldsymbol{V}$, edge-wise thresholding denoted by the function $t_\delta$ has to applied in a final step:

$$\boldsymbol{G} = \mathrm{Proj}_{h,\delta}(\boldsymbol{W}) := t_\delta(\mathrm{Proj}_h(\boldsymbol{W})) \,. \tag{20}$$

Due to the implicit correspondence between $\boldsymbol{W}$ and $\boldsymbol{G}$ over the resulting projection $\mathrm{Proj}_{h,\delta}$, the sample-efficient importance sampling as proposed in Appendix C cannot be applied. This implies a higher cost of evaluating the probability of a single DAG $\boldsymbol{G}$. Its relative frequency has to be approximated using Monte Carlo sampling:

$$q_{\mathbf{W}}(\boldsymbol{W}) = \prod_{i \neq j}^{D} q_{\boldsymbol{\phi}_{ij}}(W_{ij}) \,, \tag{21}$$

$$q_{\mathbf{G}}(\boldsymbol{G}) = \int \left[\boldsymbol{G} = \mathrm{Proj}_{h,\delta}(\boldsymbol{W})\right] q_{\mathbf{W}}(\boldsymbol{W}) \ \mathrm{d}\boldsymbol{W} = \mathbb{E}_{\mathbf{W} \sim q_{\mathbf{W}}} \left[\boldsymbol{G} = \mathrm{Proj}_{h,\delta}(\boldsymbol{W})\right] \,. \tag{22}$$

For further details on the projection, we refer the reader to Thompson et al. (2024).

## A.3 Gibbs distribution

Eq. (5) defines a generative model where the probability for any cyclic graph is negligible small given a sufficiently high prefactor $\lambda$. As in section A.1, the odds of two graphs can be evaluated even without any sampling, while the calculation of the probability of a given DAG, $q_{\mathbf{G}}(\boldsymbol{G})$ requires normalization. In contrast

to Eq. (18), for a high value of $\lambda$ samples of the generative model violate acyclicity only with a very low probability and the normalization constant $Z$ is almost 1:

$$q_{\mathbf{G}}(\boldsymbol{G}) = \frac{1}{Z} \exp\big(-\lambda h(\boldsymbol{G})\big) q_{\mathbf{A}}(\boldsymbol{G}) \qquad \text{with } Z := \sum_{\boldsymbol{G} \in \mathcal{G}} \exp\big(-\lambda h(\boldsymbol{G})\big) q_{\mathbf{A}}(\boldsymbol{G}) \left[\boldsymbol{A} \in \mathcal{G}\right] \xrightarrow[\lambda \to \infty]{} 1 \ . \qquad (23)$$

### A.4   Particle distribution

A particle distribution for DAGs as defined in Eq. (6) distributes the probability mass among a limited number of DAGs $\{\boldsymbol{G}^{(k)}\}_{k=1}^{K}$:

$$q_{\mathbf{G}}(\mathbf{G}) = \sum_{k=1}^{K} \left[\boldsymbol{G} = \boldsymbol{G}^{(k)}\right] q_{\mathrm{k}}(k) \ . \qquad (24)$$

### A.5   Mixture distribution

A mixture distribution models $K$ components of a given base distribution, e.g., RPM- or ARCO-DAG. The probability for a given DAG $\boldsymbol{G}$ then results from weighting by their individual probabilities $q_{\boldsymbol{G}^k}(\boldsymbol{G})$:

$$q_{\mathbf{G}}(\boldsymbol{G}) = \sum_{k=1}^{K} q_{\mathbf{G}^k}(\boldsymbol{G}) \, q_k(k) \ . \qquad (25)$$

### A.6   Probabilistically masked DAG

The probability of a graph $\boldsymbol{G}$ under the generative order-based model defined in Eq. (8) can be obtained by marginalization over the space of all permutations $\mathcal{P}$ over $D$ nodes and corresponding upper-triangular matrices $\mathcal{U}^{(\boldsymbol{\Pi})} \subset \mathcal{A}$, i.e., the subset of edges that is not masked by the permutation matrix $\boldsymbol{\Pi}$:

$$q_{\mathbf{G}}(\boldsymbol{G}) = \sum_{\boldsymbol{\Pi} \in \mathcal{P}, \boldsymbol{A} \in \mathcal{U}^{(\boldsymbol{\Pi})}} \left[\boldsymbol{G} = \boldsymbol{M}^{(\boldsymbol{\Pi})} \circ \boldsymbol{A}\right] q_{\boldsymbol{\Pi}}(\boldsymbol{\Pi}) \prod_{\pi(i) < \pi(j)} q_{\mathbf{A}_{ij}}(\boldsymbol{A}_{ij}) \qquad \text{with } \boldsymbol{M}^{(\boldsymbol{\Pi})} := \boldsymbol{\Pi}^T \boldsymbol{M} \boldsymbol{\Pi} \ . \qquad (26)$$

Sampling without replacement from a categorical distribution with some fixed weights $\{\psi_d\}_{d=1}^{D}$ is equivalent to drawing a permutation over $[D]$ and known as the Plackett-Luce (PL) distribution (Plackett, 1975). At each sampling stage $d$, the categorical weights for the remaining variables that were not yet sampled are normalized:

$$q_{\pi}(\boldsymbol{\pi}) = \prod_{d=1}^{D-1} q_{\pi(d)}\big(\pi(d)\big) = \prod_{d=1}^{D-1} \frac{\psi_{\pi(d)}}{\sum_{i=1}^{D} \psi_i - \sum_{j < d} \psi_{\pi(j)}} \ . \qquad (27)$$

A differentiable, continuous relaxation of the discrete sampling of permutation matrices $\boldsymbol{\Pi}$ can be obtained by pairing the Gumbel-Softmax trick (Maddison et al., 2014) with Softsort (Prillo & Eisenschlos, 2020).

### A.7   Autoregressive model over all potential edges

Denoting the mapping between the sequence of edges $\boldsymbol{S} \in \mathcal{S}$ to a graph $\boldsymbol{G}$ by $g : \mathcal{S} \mapsto \{0,1\}^{D \times D}$, the probability of sampling a graph $\boldsymbol{G}$ by the generative model defined in Eq. (10) equals:

$$q_{\mathbf{G}}(\boldsymbol{G}) = \sum_{\boldsymbol{S} \in \mathcal{S}} q_{\mathbf{G}, \mathbf{S}}(\boldsymbol{G}, \boldsymbol{S}) = \sum_{\boldsymbol{S} \in \mathcal{S}} \left[\boldsymbol{G} = g(\boldsymbol{S})\right] q_{\mathbf{S}}(\boldsymbol{S}) \ . \qquad (28)$$

where the probability of a sequence $q_{\mathbf{S}}$ is the product of the autoregressive edge probabilities and probability of stopping the generative process:

$$q_{\mathbf{S}}(\boldsymbol{S}) = \left(\prod_{i=1}^{E} q_{\mathbf{S}_i | \mathbf{S}_{:i-1}}(S_i | S_{:i-1})\right) \left(\prod_{j=1}^{E-1} q_{\mathbf{f}_j | \mathbf{S}_{:j-1}}(0 | S_{:j-1})\right) q_{\mathbf{f}_E | \mathbf{S}_{:E-1}}(1 | S_{E-1}) \ . \qquad (29)$$

The sequential process of a GFlowNet itself defines a DAG where all non-leaf nodes represent states of an unfinished causal graph $\boldsymbol{G}^{(i)} := g(\boldsymbol{S}_{:i})$. Starting with the empty graph state as the root node and probability 1, it splits the probability of a state representing a preliminary graph $\boldsymbol{G}^{(i)}$ among its children who are either other preliminary graph states with an additional edge or a terminal state, i.e. a leave node representing the probability of the DAG $\boldsymbol{G}^{(i)}$ under the GFlowNet model Deleu et al. (2022).

## B   Particle distributions

In the following we present a proof of Lemma 1 and derive the minimal values for the four statistical distances between a discrete target distribution $p$ that assigns some positive probability mass to more than $K$ graphs and a distribution $q$ with only $K$ particles. In the trivial case where $p$ distributes all probability only on $K$ or fewer graphs, a particle distribution $q$ can model $p$ perfectly without any approximation error and Lemma 1 follows directly.

### B.1   Minimal total variation distance

*Proof.* The total variation distance $D_{\text{TV}}$ between the candidate and target distribution, $q_{\mathbf{G}}$ and $p_{\mathbf{G}}$, respectively, can be rewritten as:

$$D_{\text{TV}}(q_{\mathbf{G}}\|p_{\mathbf{G}}) = \frac{1}{2} \sum_{\boldsymbol{G} \in \mathcal{G}} |q(\boldsymbol{G}) - p(\boldsymbol{G})| = \frac{1}{2} \Bigg( \underbrace{\sum_{\boldsymbol{G} \in \mathcal{G} \backslash \{\boldsymbol{G}^{(k)}\}_{k=1}^{K}} p(\boldsymbol{G})}_{= 1 - \sum_{k=1}^{K} p(\boldsymbol{G}^{(k)})} + \sum_{k=1}^{K} \left| q(\boldsymbol{G}^{(k)}) - p(\boldsymbol{G}^{(k)}) \right| \Bigg) . \tag{30}$$

In the RHS of Eq. (30), the sum over all graphs is split into two terms. The graphs that are not in the support of the particle distribution $q_{\mathbf{G}}$ lead to a distance that is independent of the individual probabilities for the $K$ graphs. The second term measures their absolute deviations which would be minimized if $p(\boldsymbol{G}^{(k)}) = q(\boldsymbol{G}^{(k)}) \, \forall k \in [K]$. However, due to the normalization constraint, the accumulated probability mass of the not modeled graphs has to be distributed additionally among the $K$ graphs in $q_{\mathbf{G}}$. Consequently, as we are interested in minimizing the divergence, $q(\boldsymbol{G}^{(k)}) \geq p(\boldsymbol{G}^{(k)}) \, \forall k \in [K]$ and thus

$$\min_{q \in \mathcal{Q}} D_{\text{TV}}(q_{\mathbf{G}}\|p_{\mathbf{G}}) = \min_{q \in \mathcal{Q}} \frac{1}{2} \left( 1 - \sum_{k=1}^{K} p(\boldsymbol{G}^{(k)}) + \sum_{k=1}^{K} \left( q(\boldsymbol{G}^{(k)}) - p(\boldsymbol{G}^{(k)}) \right) \right) = 1 - \sum_{k=1}^{K} p(\boldsymbol{G}^{(k)}) . \tag{31}$$

Consequently, both sums in Eq. (31) amount precisely to the loss of the accumulated probability mass for the graphs not modeled by $q_{\mathbf{G}}$ and the total variation distance is minimized when $q_{\mathbf{G}}$ assigns non-zero probability to the $K$ graphs with the highest probability in $p_{\mathbf{G}}$:

$$\forall k \in [K]: \quad \boldsymbol{G}^{(k)} = \operatorname*{arg\,max}_{\boldsymbol{G} \in \mathcal{G} \backslash \{\boldsymbol{G}^{(i)}\}_{i<k}} p_{\mathbf{G}}(\boldsymbol{G}) . \tag{32}$$

$\square$

Their corresponding probabilities $q(\boldsymbol{G}^{(k)})$ are not unique in general. The approximate probability distribution $q_{\mathbf{G}}$ with the minimal $D_{\text{TV}}$ to $p_{\mathbf{G}}$ does not constrain how the remaining probability mass is distributed among the $K$ modeled graphs. The probability of a single graph could account for the missing accumulated probability mass yielding only a single biased value, but distorting the relative probabilities. This would limit the utility of such an approximated distribution heavily and motivates imposing the following constraint that is stated in Lemma 1 as a premise:

$$\forall i \neq j \in [K]: \quad \frac{q(\boldsymbol{G}^{(i)})}{q(\boldsymbol{G}^{(j)})} = \frac{p(\boldsymbol{G}^{(i)})}{p(\boldsymbol{G}^{(j)})} . \tag{33}$$

Preserving the relative probabilities of $p_{\mathbf{G}}$ in $q_{\mathbf{G}}$ then uniquely defines $q_{\mathbf{G}}$ by the normalized probabilities

$$q(\boldsymbol{G}^{(k)}) = \frac{p(\boldsymbol{G}^{(k)})}{\sum_{j=1}^{K} p(\boldsymbol{G}^{(j)})} \,, \tag{34}$$

and yields an approximation that is for all $K$ graphs overconfident w.r.t. to the approximated distribution $p_{\mathbf{G}}$, i.e., $q(\boldsymbol{G}^{(k)}) > p(\boldsymbol{G}^{(k)}) \, \forall k \in [K] \subset |\mathcal{G}|$.

## B.2 Minimal Kullback-Leibler divergence

In contrast to the total variation distance, the Kullback-Leibler divergence is not symmetric. Due to the assumed smaller support of the considered particle distribution $q_{\mathbf{G}}$ compared to the target distribution over all DAGs $p_{\mathbf{G}}$ only the reverse KL divergence, $D_{\mathrm{KL}}(q_{\mathbf{G}} \| p_{\mathbf{G}})$, is well defined.

*Proof.* The constrained minimization of the KL divergence $D_{\mathrm{KL}}(q_{\mathbf{G}} \| p_{\mathbf{G}})$ can be solved by using its corresponding Lagrangian function $\mathcal{L}$ and multiplier $\gamma$:

$$\mathcal{L}(q_{\mathbf{G}}, \gamma) = \sum_{k=1}^{K} q(\boldsymbol{G}^{(k)}) \log \frac{q(\boldsymbol{G}^{(k)})}{p(\boldsymbol{G}^{(k)})} + \gamma \left( 1 - \sum_{k=1}^{K} q(\boldsymbol{G}^{(k)}) \right) \,. \tag{35}$$

Setting its derivatives w.r.t. the probability of the particles to zero, reveals that $q(\boldsymbol{G}^{(k)})$ is proportional to $p(\boldsymbol{G}^{(k)})$:

$$\frac{\partial \mathcal{L}}{\partial q(\boldsymbol{G}^{(k)})} = \log \frac{q(\boldsymbol{G}^{(k)})}{p(\boldsymbol{G}^{(k)})} + 1 - \gamma = 0$$

$$\Rightarrow q(\boldsymbol{G}^{(k)}) = e^{\gamma - 1} p(\boldsymbol{G}^{(k)}) \,. \tag{36}$$

As a direct consequence, the normalization constraint leads to the same probabilities as in Eq. (34) that preserves the relative probabilities of $p_{\mathbf{G}}$ in $q_{\mathbf{G}}$. Thus the KL divergence simplifies to:

$$\begin{aligned} D_{\mathrm{KL}}(q_{\mathbf{G}} \| p_{\mathbf{G}}) &= \sum_{\boldsymbol{G} \in \mathcal{G}} q(\boldsymbol{G}) \log \frac{q(\boldsymbol{G})}{p(\boldsymbol{G})} = \sum_{k=1}^{K} q(\boldsymbol{G}^{(k)}) \log \frac{q(\boldsymbol{G}^{(k)})}{p(\boldsymbol{G}^{(k)})} \\ &= -\sum_{k=1}^{K} q(\boldsymbol{G}^{(k)}) \log \sum_{j=1}^{K} p(\boldsymbol{G}^{(j)}) = -\log \sum_{j=1}^{K} p(\boldsymbol{G}^{(j)}) \,. \end{aligned} \tag{37}$$

Note that the negative logarithm is minimized when its argument, the sum of the probability of $K$ graphs in the target distribution $p_{\mathbf{G}}$, is maximized. Hence, the constrained KL divergence is minimized, when $q_{\mathbf{G}}$ assigns non-zero probabilities to the $K$ graphs with the highest probabilities in $p_{\mathbf{G}}$ as in Eq. (12). $\square$

## B.3 Minimal Hellinger & Bhattacharyya distance

*Proof.* The derivation of the minimal Hellinger distance $D_{\mathrm{H}}$ follows analogously to the minimal Kl divergence in section B.2. The first steps is the derivation of the constraint of preserved relative probabilities as stated in Eq. (34) from the minimization problem. Since the argument of the outer square root is non-negative and squaring is a monotonic function on $\mathbb{R}_0^+$, the distribution $q \in \mathcal{Q}$ that minimizes the Hellinger distance is preserved when it is squared:

$$\underset{q \in \mathcal{Q}}{\arg \min} \, D_{\mathrm{H}}(q_{\mathbf{G}} \| p_{\mathbf{G}}) = \underset{q \in \mathcal{Q}}{\arg \min} \, D_{\mathrm{H}}{}^2(q_{\mathbf{G}} \| p_{\mathbf{G}}) \tag{38}$$

Furthermore,

$$D_{\mathrm{H}}{}^2(q_{\mathbf{G}}\|p_{\mathbf{G}}) = \frac{1}{2}\sum_{\boldsymbol{G}\in\mathcal{G}}\left(\sqrt{q(\boldsymbol{G})}-\sqrt{p(\boldsymbol{G})}\right)^2$$

$$= \frac{1}{2}\left(\sum_{\boldsymbol{G}\in\mathcal{G}\setminus\{\boldsymbol{G}^{(k)}\}_{k=1}^K}p(\boldsymbol{G}) \;+\; \sum_{k=1}^{K}\left(\sqrt{q(\boldsymbol{G}^{(k)})}-\sqrt{p(\boldsymbol{G}^{(k)})}\right)^2\right)$$

$$= \frac{1}{2}\left(1 + \sum_{k=1}^{K}q(\boldsymbol{G}^{(k)}) - 2\sqrt{q(\boldsymbol{G}^{(k)})p(\boldsymbol{G}^{(k)})}\right)$$

$$= 1 - \sum_{k=1}^{K}\sqrt{q(\boldsymbol{G}^{(k)})p(\boldsymbol{G}^{(k)})}\,. \tag{39}$$

This allows to rewrite the minimization problem as a maximization problem:

$$\arg\min_{q\in\mathcal{Q}}D_{\mathrm{H}}{}^2(q_{\mathbf{G}}\|p_{\mathbf{G}}) = \arg\max_{q\in\mathcal{Q}}\sum_{k=1}^{K}\sqrt{q(\boldsymbol{G}^{(k)})p(\boldsymbol{G}^{(k)})}\,. \tag{40}$$

Then the corresponding Lagrangian with multiplier $\gamma$ reads:

$$\mathcal{L}(q_{\mathbf{G}},\gamma) = \sum_{k=1}^{K}\sqrt{q(\boldsymbol{G}^{(k)})p(\boldsymbol{G}^{(k)})} + \gamma\left(1 - \sum_{k=1}^{K}q(\boldsymbol{G}^{(k)})\right)\,. \tag{41}$$

Taking its derivative w.r.t $q(\boldsymbol{G}^{(k)})$, setting it to zero and squaring both sides yields:

$$\frac{\partial\mathcal{L}}{\partial q(\boldsymbol{G}^{(k)})} = \frac{1}{2}\frac{\sqrt{p(\boldsymbol{G}^{(k)})}}{\sqrt{q(\boldsymbol{G}^{(k)})}} - \gamma = 0$$

$$\Rightarrow q(\boldsymbol{G}^{(k)}) = \frac{1}{4\gamma^2}\,p(\boldsymbol{G}^{(k)})\,. \tag{42}$$

The normalization constraint then again implies Eq. (34) that can be inserted into Eq. (39) yielding the minimal squared Hellinger distance:

$$\min_{q\in\mathcal{Q}}D_{\mathrm{H}}{}^2(q_{\mathbf{G}}\|p_{\mathbf{G}}) = 1 - \sum_{k=1}^{K}\frac{p(\boldsymbol{G}^{(k)})}{\sqrt{\sum_{k=1}^{K}p(\boldsymbol{G}^{(k)})}} = 1 - \sqrt{\sum_{k=1}^{K}p(\boldsymbol{G}^{(k)})}\,. \tag{43}$$

The Bhattacharyya distance $D_{\mathrm{B}}$ is directly related to the squared Hellinger distance $D_{\mathrm{H}}^2$. Since the logarithm is a monotonic function, the previous result of the maximization problem can be reused:

$$D_{\mathrm{B}}(q_{\mathbf{G}}\|p_{\mathbf{G}}) = -\ln\left(\sum_{\boldsymbol{G}\in\mathcal{G}}\sqrt{q(\boldsymbol{G})p(\boldsymbol{G})}\right) = -\ln\left(1 - D_{\mathrm{H}}{}^2(q_{\mathbf{G}}\|p_{\mathbf{G}})\right) \tag{44}$$

$$\min_{q\in\mathcal{Q}}D_{\mathrm{B}}(q_{\mathbf{G}}\|p_{\mathbf{G}}) = -\frac{1}{2}\ln\sum_{k=1}^{K}p(\boldsymbol{G}^{(k)})\,. \tag{45}$$

Hence, the Hellinger distance and the Bhattacharyya distance are both minimized by selecting the graphs with the highest probability in $p_{\mathbf{G}}$ as in Eq. (12). $\qquad\square$

## C  Importance sampling for parametrized distributions with an auxiliary discrete structure

### C.1  Construction of the proposal distribution

Implicit generative models for discrete structures such as DAGs can define a hierarchical sampling process with discrete auxiliary variables $\mathbf{Y}$ that partition the discrete sample space. The weights of discrete objects, e.g., edges or permutations, that do not align with some target structure can be set to 0. Since the relative probabilities of admissible substructures are preserved in this case, it is equal in probability to sampling form the unconstrained distribution and rejecting any sample that is not admissible under the target structure. W.l.o.g. we assume for the remainder of the discussion that the target structure is a DAG $\boldsymbol{G}$. Denoting the unnormalized weights for a combination $(\mathbf{G}, \mathbf{Y})$ of the candidate model $q_{\boldsymbol{\phi}}$ by $w_{\mathbf{G},\mathbf{Y}}$, the proposal distribution $q_{\boldsymbol{\phi}}^*$ becomes

$$q_{\boldsymbol{\phi}}^*(\boldsymbol{G}^*, \boldsymbol{Y}^*) := \frac{w_{\boldsymbol{G}^*, \boldsymbol{Y}^*}}{\sum_{\boldsymbol{Y}} [\boldsymbol{G}^* = g(\boldsymbol{Y})] \, w_{g(\boldsymbol{Y}), \boldsymbol{Y}}} \, , \tag{46}$$

and is well defined $\forall \boldsymbol{G}^* \in \mathcal{G}$. It constrains the sample space to the target graph $\boldsymbol{G}^*$ and can be directly derived from the weights $\boldsymbol{w}$ of the candidate distribution $q_{\boldsymbol{\phi}}$ at any optimization step:

$$q_{\boldsymbol{\phi}}(\boldsymbol{G}^*, \boldsymbol{Y}^*) := \frac{w_{\boldsymbol{G}^*, \boldsymbol{Y}^*}}{\sum_{\boldsymbol{Y}} w_{g(\boldsymbol{Y}), \boldsymbol{Y}}} \, . \tag{47}$$

**Order-based model**  For order-based models as described in section 3.4 and depicted in Figure 3d, the weights of the Bernoulli distribution for any edges that do not appear in the target graph can be set to zero, i.e., $\forall i, j$ with $\boldsymbol{G}_{ij}^* = 0 : \phi_{ij} = 0$. In addition, the sampling of a total order in Eq. (27) can be constrained to the (partial) order induced by the target DAG by setting the weights of all variables to 0 as long as their parents have not been sampled as a predecessor. Note that the analytic value of $q_{\boldsymbol{\phi}}^*(\boldsymbol{G}^*, \boldsymbol{Y}^*)$ requires summation of the probabilities of all admissible total orders, in our practical implementation we approximate the sum by a finite number of sampled admissible total orders (lower bound). The more edges a DAG has, the fewer admissible permutations exists and the better the approximation by a fixed number of sampled random permutations. In the following we outline the procedure for RPM-DAG with a simple example.

**Example 3** (Collider structure). Consider the setting with $D = 3$ and assume the target graph $\boldsymbol{G}^*$ forms a collider, i.e., $\mathrm{X}_1 \to \mathrm{X}_3 \leftarrow \mathrm{X}_2$. Then two total orders with $\mathrm{X}_3$ as the last variable are admissible with $\boldsymbol{G}^*$, i.e., $\pi_1 = (123)$ and $\pi_2 = (213)$.

Under the RPM-DAG model, the probability of $\boldsymbol{G}^*$ is then the sum of the probabilities of the two options that generate $\boldsymbol{G}^*$:

$$q_{\mathbf{G}}(\boldsymbol{G}^*) = \underbrace{q_{\boldsymbol{\pi}}(\boldsymbol{\pi}_1) \, q_{\mathbf{A}_{12}}(0) q_{\mathbf{A}_{13}}(1) q_{\mathbf{A}_{23}}(1)}_{= q(\boldsymbol{G}^*, \boldsymbol{\pi}_1)} + \underbrace{q_{\boldsymbol{\pi}}(\boldsymbol{\pi}_2) \, q_{\mathbf{A}_{13}}(1) q_{\mathbf{A}_{21}}(0) q_{\mathbf{A}_{23}}(1)}_{= q(\boldsymbol{G}^*, \boldsymbol{\pi}_2)} \, . \tag{48}$$

Now consider that the proposal distribution generates $\boldsymbol{G}^*$ by sampling $\boldsymbol{\pi}_1$. Then the resulting joint probability only differs in the probability of sampling $\boldsymbol{\pi}_1$, since the not masked entries of $\boldsymbol{A}^*$ are predetermined by the entries of $\boldsymbol{G}^*$:

$$q^*(\boldsymbol{G}^*, \boldsymbol{\pi}_1) = q_{\boldsymbol{\pi}}^*(\boldsymbol{\pi}_1) q_{\mathbf{A}_{12}}(0) q_{\mathbf{A}_{13}}(1) q_{\mathbf{A}_{23}}(1) \, . \tag{49}$$

While $q_{\boldsymbol{\pi}}$ admits sampling of all possible permutations,

$$q_{\boldsymbol{\pi}}(\boldsymbol{\pi}_1) = \frac{\psi_1}{\psi_1 + \psi_2 + \psi_3} \frac{\psi_2}{\psi_2 + \psi_3} \, , \tag{50}$$

$q_{\boldsymbol{\pi}}^*$ is constrained to $\boldsymbol{\pi}_1$ and $\boldsymbol{\pi}_2$. Hence, the normalization of the weights $\boldsymbol{\psi}$ changes according to the restricted set of admissible options:

$$q_{\boldsymbol{\pi}}^*(\boldsymbol{\pi}_1) = \frac{\psi_1}{\psi_1 + \psi_2} \, . \tag{51}$$

**Sequence-based model**  Sampling of the next edge in a sequence as described in section 3.5 and depicted in Figure 3f can be directly constrained to the edges that appear in the target graph. The ancestral masking that ensures acyclicity remains unaffected by importance sampling. The number of edges in a graphs then limits the length of the sequence to consider. In contrast to order-based models, fewer edges imply fewer different sequences that generate the same DAG.

For Example 3, we sketch the implementation for GFlowNet-DAG model. Let us denote the two edges of $\boldsymbol{G}^*$ by $G_{13} \coloneqq (\mathrm{X}_1 \to X_3)$ and $G_{23} \coloneqq (\mathrm{X}_2 \to X_3)$. Then the probability of sampling $\boldsymbol{G}^*$ is the sum of both sequences $\boldsymbol{S}_{13,23} \coloneqq (G_{13}, G_{23})$ and $\boldsymbol{S}_{23,13} \coloneqq (G_{23}, G_{13})$ that generates $\boldsymbol{G}^*$:

$$q_{\mathbf{G}}(\boldsymbol{G}^*) = q_{\mathbf{G},\mathbf{S}}(\boldsymbol{G}^*, \boldsymbol{S}_{13,23}) + q_{\mathbf{G},\mathbf{S}}(\boldsymbol{G}^*, \boldsymbol{S}_{23,13}) \,. \tag{52}$$

Now assume that the sequence $\boldsymbol{S} = \boldsymbol{S}_{13,23}$ is sampled by the proposal distribution $q^*$. Starting with the empty graph, the probability of stopping has to evaluate to false, i.e., $q_\phi(\mathrm{f}_0 = 0)$. After sampling $\boldsymbol{G}_{13}$, the process continues with $\mathrm{f}_1 = 1$ and sampling $\boldsymbol{G}_{23}$, before it stops by $\mathrm{f}_2 = 1$. Note that the probabilities of discontinuation are the same for both, $q_{\mathbf{G}}$ and $q_{\mathbf{G}}^*$, since they fixed by $\boldsymbol{G}^*$. The proposal distribution only differs by renormalization of the probability for the first edge, since the second edge is the last one and there is no alternative anymore. As a consequence, the probability of sampling the sequence $\boldsymbol{S}$ to generate $\boldsymbol{G}^*$ under the proposal distribution equals

$$q_{\mathbf{G}}^*(\boldsymbol{G}^*, \boldsymbol{S}) = q_{\mathrm{S}_1}^*(\boldsymbol{S}_1) \, q_{\mathrm{f}_0}(0) \, q_{\mathrm{f}_1}(0|\{S_1\}) \, q_{\mathrm{f}_2}(1|\{S_1, S_2\}) \,. \tag{53}$$

## C.2 Sample-efficiency of the proposed importance sampling

In the following, we provide the proof of Lemma 2 by computing the effective sample size Kong (1992):

*Proof.* Let's define $\hat{q}_{\mathrm{MC}}$ as the standard Monte Carlo estimator of $q_\phi(\boldsymbol{G}^*)$ based on Eq. (15):

$$\hat{q}_{\mathrm{MC}}(\boldsymbol{G}^*) = \frac{1}{N} \sum_{n=1}^{N} \left[ \boldsymbol{G}^* = g(\mathbf{Y}^{(n)}) \right] \,, \tag{54}$$

and $\hat{q}_{\mathrm{IS}}$ as the importance sampling estimator based on Eq. (16) with $(\mathbf{G}^{*(n)}, \mathbf{Y}^{*(n)})$ as samples from $q_\phi^*$:

$$\hat{q}_{\mathrm{IS}}(\boldsymbol{G}^*) = \frac{1}{N} \sum_{n=1}^{N} r_\phi \left( \boldsymbol{G}^{*(n)}, \mathbf{Y}^{*(n)} \right), \quad \text{where } r_\phi(\boldsymbol{G}^*, \mathbf{Y}^*) \coloneqq \frac{q_\phi(\boldsymbol{G}^*, \mathbf{Y}^*)}{q_\phi^*(\boldsymbol{G}^*, \mathbf{Y}^*)} \tag{55}$$

is the likelihood ratio of the target and proposal distribution that is well-defined by construction $\forall \boldsymbol{G} \in \mathcal{G}$. Both estimators are unbiased, but have different variances:

$$\begin{aligned}
N\mathrm{Var}\big[\hat{q}_{\mathrm{MC}}(\boldsymbol{G}^*)\big] &= \mathbb{E}_{\mathbf{Y}\sim q_\phi}\left[ \left[\boldsymbol{G}^* = g(\mathbf{Y})\right]^2 \right] - \mathbb{E}_{\mathbf{Y}\sim q_\phi}\left[ \left[\boldsymbol{G}^* = g(\mathbf{Y})\right] \right]^2 \\
&= \mathbb{E}_{\mathbf{Y}\sim q_\phi}\left[ \left[\boldsymbol{G}^* = g(\mathbf{Y})\right] \right] - q_\phi(\boldsymbol{G}^*)^2 \\
&= q_\phi(\boldsymbol{G}^*) - q_\phi(\boldsymbol{G}^*)^2 \,, \\
N\mathrm{Var}\big[\hat{q}_{\mathrm{IS}}(\boldsymbol{G}^*)\big] &= \mathbb{E}_{\mathbf{Y}^*\sim q_\phi^*}\left[ r_\phi\left(\boldsymbol{G}^*, \mathbf{Y}^*\right)^2 \right] - \mathbb{E}_{\mathbf{Y}^*\sim q_\phi^*}\left[ r_\phi\left(\boldsymbol{G}^*, \mathbf{Y}^*\right) \right]^2 \\
&= \sum_{\boldsymbol{Y}^*} q_\phi(\boldsymbol{G}^*, \mathbf{Y}^*) \, r_\phi\left(\boldsymbol{G}^*, \mathbf{Y}^*\right) - q_\phi(\boldsymbol{G}^*)^2 \\
&= \mathbb{E}_{\mathbf{Y}\sim q_\phi}\left[ r_\phi(\mathbf{G}, \mathbf{Y}) \left[\boldsymbol{G}^* = g(\mathbf{Y})\right] \right] - q_\phi(\boldsymbol{G}^*)^2 \,.
\end{aligned} \tag{56}$$
$$\tag{57}$$

The derivation of Eq. (57) switches the reference distribution for the expectation value from $q_\phi^*$ to $q_\phi$ and is a purely notational transformation. Note that the summation is still limited to $(\boldsymbol{G}^*, \boldsymbol{Y}^*)$ which explains the added Iverson bracket. Since the parameters $\boldsymbol{\phi}$ are shared and $q_\phi(\boldsymbol{G}^*)$ is computed by renormalization over

the constrained space of $\mathbf{Y}^*$, we have $\forall\, \boldsymbol{G}^* \in \mathcal{G} : r_\phi(\boldsymbol{G}^*, \mathbf{Y}^*) < 1$ and, hence, it is trivial that the effected sample size has to be strictly greater than N:

$$\mathrm{ESS}\big(\hat{q}_{\mathrm{IS}}(\boldsymbol{G}^*)\big) \coloneqq N \frac{\mathrm{Var}\big[\hat{q}_{\mathrm{MC}}(\boldsymbol{G}^*)\big]}{\mathrm{Var}\big[\hat{q}_{\mathrm{IS}}(\boldsymbol{G}^*)\big]} = \frac{1 - q_\phi(\boldsymbol{G}^*)}{\frac{\mathbb{E}_{\phi^*}\big[r_\phi(\boldsymbol{G}^*, \mathbf{Y})\big]}{q_\phi(\boldsymbol{G}^*)} - q_\phi(\boldsymbol{G}^*)} N > N \ . \tag{58}$$

$$\square$$

Whenever there exists only one admissible $Y^*$ that generates $\boldsymbol{G}^*$, e.g., for RPM-DAG when $\boldsymbol{G}^*$ implies a total order or for the GflowNet-DAG when $\boldsymbol{G}^*$ is the empty graph, the likelihood ratio $r(\boldsymbol{G}^*, \boldsymbol{Y}^*)$ is constant and equals the very probability of sampling $\boldsymbol{G}^*$:

$$r_\phi(\boldsymbol{G}^*, \boldsymbol{Y}^*) = \frac{\sum_{\boldsymbol{Y}}[\boldsymbol{G}^* = g(\boldsymbol{Y})]\, w_{g(\boldsymbol{Y}), \boldsymbol{Y}}}{\sum_{\boldsymbol{Y}} w_{g(\boldsymbol{Y}), \boldsymbol{Y}}} = \sum_{\boldsymbol{Y}^*} \frac{w_{g(\boldsymbol{Y}^*), \boldsymbol{Y}^*}}{\sum_{\boldsymbol{Y}} w_{g(\boldsymbol{Y}), \boldsymbol{Y}}} = \sum_{\boldsymbol{Y}_*} q_\phi(\boldsymbol{G}^*, \boldsymbol{Y}^*) = q_\phi(\boldsymbol{G}^*) \ . \tag{59}$$

This directly implies then that $\mathrm{Var}[\hat{q}_{\mathrm{IS}}(\boldsymbol{G}^*)] = 0$ as a single sample of the proposed importance sampling estimator yields the true value. In this case, the estimator *Eq.* (55) is then optimal w.r.t. to sample-efficiency. In the general case, the proposal distribution in Eq. (46) is not optimal, but sample-efficient according to Eq. (58).

## D  Details on experimental setup

### D.1  Model architectures

**RPM-DAG**  Following the described parametrization of DPM-DAG, we model an off-diagonal adjacency matrix containing the logits of Bernoulli distributions for the directed edges and log-weights for the PL distributions. In contrast to DPM-DAG, we implemented discrete sampling of permutation and adjacency matrices and no continuous relaxations of it.

**ARCO-DAG**  The autoregressive neural network of ARCO-DAG consists of a simple two layer perceptron with $H_N = 30$ hidden neurons and ReLU-activations and follows the official implementation on `https://github.com/chritoth/bci-arco-gp/`.

**GFlowNet-Dag**  The transformer architecture for the GFlowNet-Dag model follows the official implementation provided on `https://github.com/tristandeleu/jax-dag-gflownet`, as default parameters it uses an embedding size of $H_E = 128$, a key size of $H_K = 64$ and $H_L = 7$ layers of Transformer blocks.

### D.2  Experimental parameters

In the experiments in section 5.1 and 5.2, we evaluate the forward KL divergence in all three, respectfully four, graphs with positive probability mass. The probability of a graph under the candidate model is approximated by Eq. (16) using *importance samples* (IS). To speed up the calculations during training, the number of IS is chosen to be smaller than for the final evaluation of the trained model. The learning rate was chosen by a grid search over $\{1, 5 \times 10^{-1}, 1 \times 10^{-1}, 5 \times 10^{-2}, 1 \times 10^{-2}, 5 \times 10^{-3}, 1 \times 10^{-3}, 5 \times 10^{-4}, 1 \times 10^{-4}\}$. Training was performed over 1000 optimization steps to account for the instability of GFlowNet-DAG. We document the used hyperparameters in Table 5.

Due to the low number of DAGs, our first experiment (section 5.1) allows to evaluate all 25 graphs within a single forward pass during training and, hence, one optimization step equals one epoch. To account for the higher number of possible sequences, we average the probability of sampling a target graph from the model distribution using 10 importance samples for GFlowNet-DAG. For evaluation of each graph probability we drew 100 importance samples for all methods.

In the second experiment (section 5.2), we had to limit the number of samples from the target distribution for a single optimization step. Due to the high number of trainable parameters we reduced the number of samples from the target distribution from 100 to 25 and used 5 instead of 10 importance samples for GFlowNet-DAG.

For the evaluation of each of the 543 DAGs, we decreased the number of importance samples from 100 to 20.

Table 5: Hyperparameters.

(a) For the experiments in section 5.1 and 5.2.

| Graph model | Learning rate | # forward KL samples | # IS for training | # IS for evaluation |
|---|---|---|---|---|
| RPM-DAG | 0.5 | 25 | 1 | 100 |
| ARCO-DAG | 0.5 | 25 | 1 | 100 |
| GFlowNet-DAG | 0.001 | 25 | 10 | 100 |

(b) For the experiments in section 5.3.

| Graph model | Learning rate | # forward KL samples | # IS for training | # IS for evaluation |
|---|---|---|---|---|
| RPM-DAG | 0.05 | 100 | 10 | 20 |
| ARCO-DAG | 0.05 | 100 | 10 | 20 |
| GFlowNet-DAG | 0.001 | 25 | 5 | 20 |

## D.3 Training time

In Table 6 we provide a qualitative comparison of measured training times of the different graph models for the experiment described in section 5.3 with the hyperparameters listed in Table 5. All models were evaluated over 20 independent runs, each consisting of 1000 optimization steps. The computations were conducted on a 11th Gen. Intel(R) Core™ i7-1165G7 processor with 2.80 GHz, 4 cores and 8 logical processing units paired with 32 GB of DDR SDRAM. The reported average times for RPM- and ARCO-DAG scale linearly with the number of mixture components, while GlowNet-DAG takes more than 250 times longer than their base models. Its increased computation time is a result of the huge number of parameters of the transformer architecture and the sequential generation of the DAGs.

Table 6: Averaged training time and standard deviations over 20 runs for the probabilistic graph models in the experiment of section 5.3 alongside the number of the models' parameters.

| Graph model | # learnable parameters | Training time in min |
|---|---|---|
| RPM-DAG | 100 | $0.152 \pm 0.005$ |
| 5-RPM-DAG | 505 | $0.761 \pm 0.018$ |
| 10-RPM-DAG | 1 010 | $1.51 \pm 0.04$ |
| 20-RPM-DAG | 2 020 | $3.13 \pm 0.16$ |
| ARCO-DAG | 646 | $0.181 \pm 0.004$ |
| 5-ARCO-DAG | 3 235 | $1.855 \pm 0.085$ |
| 10-ARCO-DAG | 6 470 | $1.79 \pm 0.05$ |
| 20-ARCO-DAG | 12 940 | $3.70 \pm 0.03$ |
| GFlowNet-DAG | 6 176 722 | $46.4 \pm 3.4$ |

