# OpenReview forum: "Expressiveness of Parametrized Distributions over DAGs for Causal Discovery"
_TMLR — Accepted by TMLR_

### Review · Reviewer_sSDn · 2025-07-01

**Summary Of Contributions:**

The authors present a detailed review of probabilistic approaches for representation of directed acyclic graphs. In their experiments, they compare an assortment of recently contributed probabilistic representations and assess them via forward KL divergence and total variational distance. Due to the small enumeration of graphs in the evaluation (where the four node setup yields 543 possible DAGs) a complete evaluation over the entire enumeration is possible. The focus of the evaluation is to assess the expressiveness of each probabilistic representation considered.

**Audience:**

Yes

**Broader Impact Concerns:**

No ethical or broader impact concerns.

**Claims And Evidence:**

Yes

**Requested Changes:**

In addition to the notes in the Strengths and Weaknesses section above, the following changes should be considered:

Section 2 Preliminaries, paragraph 1 probability and random variables. Line 6, the sentence "We mainly apply this concise notation in the main text" does not read well: "mainly" is superfluous.

Section 2 Paragraph: Preliminaries, functional causal models. As this is a review, a figure would be well-placed to support the notation and concepts.

Section 2 Bayesian causal discovery: the marginal likelihood arises by "averaging over the (conditional) prior distribution" is ambiguous. Changing "averaging" to "taking the expectation over" may help.

**Strengths And Weaknesses:**

The paper considers several approaches for representation that already exist in the literature, and conducts a comprehensive evaluation on a single synthetic problem set.

The authors use clear explanations and notation in their exposition. The concepts and complications of probabilistic DAG representation are clearly explained, as are the evaluation methods.

While mostly complete, there are some suggestions below which should be considered by the authors.

## Section 2, paragraph Bayesian causal discovery

The notation $q(\mathcal{D}|\mathbf{G})p(\mathbf{G})$ is introduced in the Bayesian causal discovery paragraph. There are two things to note: (1) This notation for a generative model should be introduced earlier in the Probability and random variables paragraph. (2) $q(\mathcal{D}|\mathbf{G})p(\mathbf{G})$ is the product of a generative $q$ and the prior $p(\mathbf{G})$, however in the subsequent paragraph on Bayesian model error line 5, it states "The model for the prior distribution $q(\mathbf{G})$...". This is an ambiguous switch of notation without clarification.

## Section 3.1 Independent edges

Another recent work that proposes the projection of a cyclic $\mathbf{A}$ onto a non-cyclic set as the constraint function $h(\mathbf{G})$ is Thompson et al (2024).

## Section 3.3 Probabilistic models over orders

The permutation-upper-triangular representation has been discussed previously in the literature but these citations are missing. For instance, distributions over permutations (Stern 1990), and recent (preprint) work exploring distributions over this representation (Bonilla et al 2024, Davies et al 2025). In particular, an all-at-once fully-autoregressive representation over $\mathbf{\Pi},\mathbf{U}$ was explored in Davies et al 2025 where an extension to the MADE (Germain et al, 2015) architecture was used to represent a fixed-size distribution over this tuple.

Equation 8 in section 3.3 repeats much of equation 7, section 3.2. The split of the introduction and discussion permutation-upper-triangular representation over two sections should be revisited in order to reduce ambiguity. Especially since, as stated in line 10 of Section 3.2, the particle representation in general can be defined as a probabilistic representation.

## Sections 4/5 Evaluation/Experiments

Relying solely on the structural Hamming distance for evaluation of gthe concentration of probability mass (Table 3) can be misleading. For a more comprehensive evaluation, can the authors please also provide results using other standard scores such as F1 and/or AUROC? It might also be worth referencing or adding a discussion on why use of a single score such as SHD can be misleading.

It appears that the synthetic data used for evaluation is generated from a four-node DAG. Hence, the scalability of each probabilistic representation is not assessed in higher-node scenarios, a topic of recent interest in the research community. Perhaps this restriction could be highlighted to avoid ambiguity.

## References:

Thompson, R., Bonilla, E. V., & Kohn, R. (2024). ProDAG: Projection-Induced Variational Inference for Directed Acyclic Graphs. arXiv preprint arXiv:2405.15167.

Hal Stern. Models for distributions on permutations. Journal of the American Statistical Association, 85(410):558–564, 1990.

Bonilla, E. V., Elinas, P., Zhao, H., Filippone, M., Kitsios, V., & O'Kane, T. (2024). Variational dag estimation via state augmentation with stochastic permutations. arXiv preprint arXiv:2402.02644.

Davies, L., Mackinlay, D., Oliveira, R., & Sisson, S. A. (2025). Amortized variational transdimensional inference. arXiv preprint arXiv:2506.04749.

Germain, M., Gregor, K., Murray, I., & Larochelle, H. (2015, June). Made: Masked autoencoder for distribution estimation. In International conference on machine learning (pp. 881-889). PMLR.

---

> ### Author Response · Authors · 2025-08-10
> **Rebuttal 1/3**
>
> *Thank you for reviewing our work. Please find below our responses to your comments, questions, and requested changes.*
>
>
> **§2 Preliminaries** (Bayesian causal discovery)
>
> 1) Our section on the fundamentals introduces the necessary concepts stepwise and is well-structured in three paragraphs. The full data generating process is introduced in Eq. 1 where it appears naturally within the Bayesian framework. The preceding two paragraphs do not yet consider any distribution over graphs. Therefore, we prefer to keep the notation for a generative model where it is.
>
>
>  2) The generative model for the prior is in general only an approximation, as clarified in the paragraph on Bayesian model error. Hence, having corrected the typo, it should read:
> > (...) the joint distribution $p(\mathbf{G},\mathcal{D})$ is approximated by a generative model $q(\mathcal{D}|\mathbf{G})q(\mathbf{G})$ (...).

---

> ### Author Response · Authors · 2025-08-10
> **Rebuttal 2/3**
>
> **§ 3.1 Independent Edges**
>
> 3) Thank you for the reference of Thompson et. al (2024). We did not know their (still unpublished) work. Nevertheless, we consider it an interesting contribution on the research on distributions over DAGs, cited it and added a discussion of their work in the main text.
> They propose to project samples of an unconstrained, weighted adjacency matrix by iterative L2 projection on the space of acyclic adjacency matrices and further apply L1 sparsity regularization.
> However, their proposed generative model does not allow to evaluate the probability of a target graph other than pure sampling (Eq. 14). Projecting samples of weighted adjacency matrices to the space of DAGs does not partition the space of DAGs which would allow to define an efficient proposal distribution. Consequently, the proposed evaluation using importance samples (Eq. 15) cannot be applied and supervised training with samples from the target distribution or evaluating a learned posterior is very inefficient.
> While Thompson et al. (2024) explicitly ”emphasize that (their implicit) representation (of the resulting distribution) is purely conceptual” (section 3.1), it hints at a connection to permutation-based approaches that constrain an unconstrained adjacency matrix by an acyclicity mask. After the projection on the space of weighted adjacency matrices, the“sparsity pattern $\mathbf{S} \in {0,1}^{D \times D}$ (i.e., a binary  acyclic adjacency matrix)” corresponds to a permutation. To prevent sampling a fully connected DAG, thresholding or sparsity regularization of the entries of the adjacency matrix would have to applied.
> In addition to the integration in the main text, we added the following subsection to Appendix A:
>
>     > $\textbf{Projection of samples from an unconstrained, continuous adjacency matrix}$\
>     > To avoid costly rejection sampling in the generative process, a continuous adjacency $\mathbf{W}$ drawn form the generative model defined in Eq. 4 can be projected on the space of DAGs $\mathcal{G}$, e.g., using a non-negative constraint function $h(\mathbf{W})$.
>     > This comes at the high cost for evaluating the probability of a single DAG $\mathbf{G}$.
>     > Due to the unclear correspondence between $\mathbf{W}$ and $\mathbf{G}$ over the projection $\mathrm{Proj}_{h, \delta}$, importance sampling as proposed in section 4.2 cannot be applied.
>    > The relative frequency has to be approximated using Monte Carlo sampling:
>
>    $q_\mathbf{W}(\mathbf{W}) = \prod_{i \neq j}^D{ q_{\phi_{ij}}(\mathbf{W}_{ij}) }$
>
>    $q_\mathbf{G}(\mathbf{G}) =  \mathbb{E}[ \mathbb{I}[\mathbf{G} = \text{Proj}_{h, \delta}(\mathbf{W})] ]$
>
>
>
> **§ 3.3 Probabilistic models over orders**
> > The permutation-upper-triangular representation has been discussed previously in the literature, but these citations are missing.
>
> 4) We begin this section with citing the seminal work by Teyssier and Koller (2005), crediting them for applying the idea of an order-based search in causal discovery. Subsequently, we cite peer-reviewed works on probabilistic models over permutations for causal discovery.
>
>     Inevitably, we may not have covered the most recent works that have not been published with peer review yet. Since we did not find explicit statements in the TMLR review guidelines on comparisons with very recent work, we followed the guidelines issued by [ICLR](https://www.iclr.cc/Conferences/2025/ReviewerGuide).
>
>     As pointed out by the reviewer, the three listed works, Thompson et al. (2024), Bonilla et al. (2024) & Davies (2025), are preprints published without peer-review:
>     - Davies et al. (2025) was only uploaded to arXiv *after* the submission of our work and could not be considered. Hence, we omit it for the rebuttal.
>     - Bonilla et al. (2024) seems very close to the cited prior work by Annadani et al. (2023) and Rittel & Tschiatschek (2023) that both have been published with peer-review before. According to Bonilla et al. (2024), the later only differs in evaluating the KL divergence over the permutation but shares the idea of using weights for each node (c.f. their section on related work and Appendix B).
>     In this work, we did not use Gumbel-Softmax, but score-function gradient estimation for stage-wise categorical sampling. To emphasize this difference to DPM-DAG, we introduced the abbreviation RPM-DAG in section 5.
>     - As outlined, we added a discussion for the work by Thomposon et al. (2024).
>     - The citations of Stern (1990) and Germain et al. (2015) only provide the necessary background for Bonilla et al. (2024) and Davies et al. (2025).

---

> ### Author Response · Authors · 2025-08-10
> **Rebuttal 3/3**
>
> **§ 5. Experiments**
>
> > Relying solely on the structural Hamming distance for evaluation of the concentration of probability mass (Table 3) can be misleading. For a more comprehensive evaluation, can the authors please also provide results using other standard scores such as F1 and/or AUROC? It might also be worth referencing or adding a discussion on why use of a single score such as SHD can be misleading.
>
> 5. We compare the expressiveness of candidate distributions over DAGs to match target distributions in a supervised setting using statistical divergences and focussed in the evaluation on the total variation distance and KL divergence. Using point metrics as SHD, F1 or AUROC would miss the point of our investigation.\
> The caption of Table 3 transparently communicates that it’s a target distribution. Its generation using the SHD is explained in the second paragraph of section 3.3.
>
>
> > It appears that the synthetic data used for evaluation is generated from a four-node DAG. Hence, the scalability of each probabilistic representation is not assessed in higher-node scenarios, a topic of recent interest in the research community. Perhaps this restriction could be highlighted to avoid ambiguity.
>
> 6. In the last paragraph of section 5.1 we reasoned that the investigated small graphs appear as subgraphs of bigger graphs. Any approximations error for the probability for them, transfer to bigger graphs that contain them. Hence, they are a key building pattern that justifies the focus on them.
> We are transparent in the presentation of our findings for which we provide theoretical arguments and/or experimental evidence. Since we did not test the candidate distributions on a high or moderate number of variables, we carefully formulated the last sentence of in section 6 as a conjecture. For completeness, we repeat here our reasoning for it:
> W.r.t to the scalability of the candidate distributions we point to Table 1 and the obtained results for the GFlowNet-DAG model. In our conclusion, we show that while
> GFlowNet-DAG is theoretically the most expressive candidate model, in the experiment described in section 5.3 it is easily outperformed by RPM-DAG, ARCO-DAG and even a particle representation with only 50 graphs.
> Yet it contains by far the most parameters (Appendix B list the used embedding size of $H_E = 128$, key size of $H_K = 64$ and $H_L = 7$ layers of Transformer blocks).
>
> **(Explicit) requested changes**
>
> 7) We agree with the reviewer and will remove the following sentence completely, since it is indeed not needed:
> > We mainly apply this concise notation in the main text to introduce the generative models.
>
> 8) The focus of our work are distributions over DAGs and not functional causal models. For the two simple examples 1 and 2 we visualized all graphs on which the probability mass should be distributed.
>
> 9) By using the term averaging, we alluded here to Bayesian model averaging.
> To further enhance clarity, we will reformulate the sentence as follows:
> > The marginal likelihood $p(\mathcal{D} | \mathbf{G})$ arises by averaging $(\mathcal{D}, \Theta) $ over the (conditional) prior distribution of model parameters $p(\theta | \mathbf{G})$.

---

### Review · Reviewer_G1Wi · 2025-07-02

**Summary Of Contributions:**

The paper reviews Bayesian causal structure learning algorithms to uncover their limitations. It analyzes the expressiveness of distributions over directed acyclic graphs obtained in these methods and verifies the findings experimentally. Specifically, the preprint considers five model types. It investigates how they perform on three example problems, which aim to assess if algorithms can learn correctly in certain basic cases, such as when one considers Markov equivalent graphs or when one variable depends on two variables simultaneously. Based on the results, the preprint concludes that mixtures of RPM-DAG and ARCO-DAG models are the most expressive.

**Audience:**

Yes

**Broader Impact Concerns:**

None.

**Claims And Evidence:**

Yes

**Requested Changes:**

**Crucial**

C1. Currently, while testing for the fundamental issues, the examples are small, with either 3 or 4 variables and only one instance of each problem.
Before suggesting changes, I would appreciate clarification of the authors' perspective, specifically on whether it would be feasible to add experiments with several graphs and more variables in each example and whether additional experiments would add value to the results.
Based on my understanding, additional experiments can help better assess methods for which evaluation results are not deterministic, such as GFlowNet-DAG, and also reason more confidently about how well the methods can scale.

**Minor**

M1. Add a definition of expressiveness somewhere at the beginning of the paper, along with an explanation of the approach used to measure it. Expressiveness often appears in varied contexts with different definitions, and works use different methods to measure it. Hence, a clear definition would help with the understanding.

M2. Improve the consistency of the discussion of the suggested mixture models.

 (a) Add a subsection defining proposed mixtures and reference it in the Contributions paragraph since there is no dedicated section introducing them, and the explanation is fragmented.

 (b) Specify the models in the mixtures in the last sentence of the Abstract, "To overcome them, we propose mixture models of distributions over DAGs," since otherwise, it suggests that an entirely new model is introduced.

 (c) Rephrase the last sentence in the conclusion, "Therefore, we conjecture that a mixture of RPM-DAG models is particularly suited to scale Bayesian causal discovery to higher numbers of variables." which is not well-supported by the current experimental setup, given that the experiments are for a single instance of each problem and only for a small number of variables.

M3. Add a summary statement or a table summarizing the method limitations since, currently, explanations are distributed across multiple sections.

**Strengths And Weaknesses:**

**Strengths**

S1.
The paper provides an instructive summary of recent Bayesian causal structure learning approaches. The review is well-written and easy to follow.

S2. The submission identifies three illustrative problems that can provide insight into the fundamental limitations of causal structure learning algorithms and uses them to analyze the behavior of existing methods.

S3.
The paper discovers that using mixtures of RPM-DAG and ARCO-DAG instead of individual models allows them to overcome their shortcomings in expressiveness.

**Weaknesses**

W1. The problems used to evaluate the algorithms are small in terms of the number of variables: two have three variables, and one has four. Furthermore, there is only one instance of each problem. How well the conclusions would scale to cases with more variables is unclear.

W2. The paper's organization is slightly confusing. For instance, there is no clear definition of expressiveness or summary of the method limitations, and an explanation of the proposed mixture models is fragmented.

*Kindly note that causal structure discovery is out of my area of expertise, so I cannot confidently assess the relevance or novelty of the results. I checked the general logic and presentation of the preprint.*

---

> ### Author Response · Authors · 2025-08-10
> **Rebuttal 1/2**
>
> *We thank the reviewer for their thoughtful comments.*
>
> \
> **Crucial requested changes:**
> \
> \
> **C1:** Note that the number of DAGs is superexponential in D (OEIS Founda-
> tion Inc., 2023):
>
> | D | Number of DAGs with D nodes |
> | --: | ---------: |
> |  1 | 1 |
> |  2 | 3 |
> |  3 | 25 |
> |  4 | 543 |
> |  5 | 29281 |
> |  6 | 3781503 |
> |  7 | 1138779265 |
> |  8 | 783702329343 |
> |  9 | 1213442454842881 |
> |10 | 4175098976430598143 |
>
> For the considered supervised setting we have to store a complete target distribution. This involves enumeration of all graphs and assigning them a probability. Moreover, to evaluate the learned implicit distributions we estimate their individual probabilities using importance samples. Even for a rather small number of graphs this is computationally very demanding, e.g., increasing $D$ from $4$ to $5$ yields 50 times the number of graphs
>
>  In our analysis in section 3, we identified two theoretical limitations for the candidate distribution, namely:
>  - splitting the probability mass equally among all graphs of the same Markov equivalence class, MEC (example 1),
>  - dealing with coupled edges (example 2).
>
> Based on these insights, we designed the experiments in section 5.1 and 5.2 to validate them empirically. In addition, we tested the synthetic graph distribution in Table 3 that concentrates its probability mass on graphs around the assumed underlying true graph depicted in Figure 5. Please note that this graph does not favor any of the candidate distributions.
>
> We agree that scaling the experiments to higher numbers would further strengthen the experimental evidence. However, the computational complexity of the problem sets clear limitations and it is questionable whether the scaling provides additional insights. Mixtures of k distributions are theoretically more expressive than distributions of k particles and was confirmed in the third experiment with $D=4$. We do not have any reason to believe that for graphs with a higher number of variables this changes.

---

> > ### Author Response · Authors · 2025-08-10
> > **Rebuttal 2/2**
> >
> > **Minor requested changes:**
> >
> > - **M1:**\
> > As suggested, we added the following definition of expressiveness:
> >
> >     > **Def. 1 [Expressiveness of a distribution]**\
> >     > Let $\mathcal{Q}$ be a family of distributions parametrized by $\phi$ and $p$ some target distribution over the same sample space $\Omega$.  The expressiveness of this family $\mathcal{Q}$ is defined as the ability to approximate the distribution $p$ w.r.t. to some statistical divergence or distance measure $D$, i.e. $\min_{q_\phi \in \mathcal{Q}} D(q_\phi \| p)$.
> >
> >     In the last paragraph on section 2, the beginning of section 3 as well as the supporting figure 1, we motivated approximations to a full distribution that enumerates all DAGs and assigns them some probability mass. Therefore, we place the definition above after the first paragraph of section 3, before reviewing the candidate models and their theoretical limitations. In section 4. on *Evaluation*, we provide details on the empirical evaluation and used statistical distances and divergences.
> >
> >
> > - **M2:** *Consistency of the discussion of the suggested mixture models*
> >
> >     a) Mixture models are a well-known concept in probabilistic modelling that uses a categorical distribution of base models.
> >     We provide an illustrative example in section 3.2 and motivate the distinction to the used term of a particle distribution.
> >     The base models, RPM- and ARCO-DAG, are well described in section 3.3 and Appendix A. Their mixtures are introduced at the beginning of section 5 and illustrated in Figure 3d.
> >     To enhance the presentation, we added a direct note on mixtures of order-based models at the end of section 3.4 and the following subsection to Appendix A:
> >
> >     > **General mixture distribution**\
> >     > A general mixture distribution models $K$ instances of a given base distribution, e.g., RPM- or ARCO-DAG. The probability for  a given DAG $\mathbf{G}$ then results from weighting by their individual probabilities $q_{\mathbf{G}^{k}}(\mathbf{G})$:\
> >    > $\displaystyle q_\mathbf{G}(\mathbf{G}) = \sum_{k=1}^K{ q_{\mathbf{G}^{k}}(\mathbf{G})  q_k(k)}$
> >
> >     b) To the best of our knowledge, the idea of using mixture models of distributions over DAGs has not been proposed or investigated before, although the general concept is well-known inside the probabilistic ML community and its application straight-forward. We did not imply that we designed a new base distribution, but propose mixtures of the considered models to overcome the outlined limitations. Nevertheless, we carefully considered this feedback and highlighted this point in our revision of the paper to enhance clarity on the matter.
> >
> >
> >     c) We agree with the reviewer that our work does not provide sufficient experimental evidence that would allow for a clear conclusion on the performance of candidate distributions for graphs with higher number of variables, consequently we carefully stated it as a conjecture and not as a claim.\
> >    For completeness, we summarize our reasoning in the conclusion here:
> >      - RPM-DAG is the model which support is limited to DAGs and has the smallest number of learnable parameter (Table 1),
> >      - in the experiment described in section 5.3 it outperforms a GFlowNet-DAG model that uses a magnitude of parameters (Appendix B list the used embedding size of $H_E = 128$, key size of $H_K = 64$ and $H_L = 7$ layers of Transformer blocks)
> >      - Mixtures of RPM-DAG outperforms a mixture of ARCO-DAG with the same number of base models in all our experiments, although a single ARCO-DAG model is more expressive than a single RPM-DAG model
> >
> >
> > - **M3:**\
> > In the last paragraph of section 1 we provide a summary of our contribution that lists the investigated limitations.\
> > For the considered candidate models, we discuss in sections 5.1, 5.2 and 5.3 their empirical performance and contrast them with the theoretical analysis from sections 3.2, 3.3 and 3.4.

---

### Review · Reviewer_4dVp · 2025-07-27

**Summary Of Contributions:**

The authors of this work discuss the expressivity of different parametrized distributions over DAGs for causal discovery. Specifically they focus their attention on learning the structure of a Functional Causal Model under the assumption of causal sufficiency.

The distribution considered by the authors are:

- Independent edges
- Graph particles
- DPM-DAG
- ARCO-DAG
- GFlowNet-DAG

Furthermore, the authors propose to use mixture models of DPM-DAG and ARCO-DAG.

Finally, the paper presents a set of synthetic experiments to compare per performances of the
models in terms of statistical divergence between the learned model and the real one.

**Audience:**

Yes

**Broader Impact Concerns:**

No Concerns

**Claims And Evidence:**

Yes

**Requested Changes:**

1.  I suggest dedicating an entire section to the mixture model approach in order to make more clear the separation between well known models and the proposed mixture model.
2. I suggest to briefly discuss the execution time of the different approaches.
3. Do you think that an extension of this work to the classical Bayesian network with discrete variables would be possible/useful as future work?

**Strengths And Weaknesses:**

## Strengths

This works effectively summarizes different causal structure learning approaches, discussing pros&cons of each algorithm and comparing them with a set of synthetic experiments. Moreover, an extension for DPM-DAG and ARCO-DAG is presented.

## Weakness

The authors clearly explain why they decided to use only synthetic data, and the provided explanation is definitely reasonable. Nevertheless, the presence of experiments with real world data would have allowed the reader to understand the behavior of the different models in a non-synthetic scenario.

Furthermore, the authors did not explicitly discuss the time complexity of the different models.

---

> ### Author Response · Authors · 2025-08-10
> **Rebuttal**
>
> *We thank the reviewer for their service as a reviewer and respond to the requested changes below:*
>
>
> **C1:** *Adding a dedicated section to the mixture approach for clarification*\
> In the main text, in section 3.3 on “Particle representations & mixture models”,  we highlight the difference between the two model types and set the motivation for modeling also a distribution over permutations, not only the entries of an off-diagonal adjacency matrix.
> To enhance the presentation as suggested, we added a direct note on mixtures of order-based models at the end of section 3.4 and the following subsection to Appendix A:
>
> > **General mixture distribution**\
> A general mixture distribution models $K$ components of a given base distribution, e.g., RPM- or ARCO-DAG. The probability for a given DAG $\mathbf{G}$ then results from weighting by their individual probabilities $q_{\mathbf{G}^{k}}(\mathbf{G})$:
> $\displaystyle q_\mathbf{G}(\mathbf{G}) = \sum_{k=1}^K{ q_{\mathbf{G}^{k}}(\mathbf{G}) \, q_k(k)}$
>
>
> **C2:** *Brief discussion of the execution times*\
> We will add a table showing run times for the tested distributions.
>
>
> **C3:** *Assessment of an extension of this work to Bayesian networks with discrete variables*\
> We are not sure if we fully understood the question and would ask the reviewer for further clarification.
> In this work, we did not make an assumption about the data type for the endogenous variables of the functional causal model that defines a Bayesian network.

---

### Review · Reviewer_jnvY · 2025-07-29

**Summary Of Contributions:**

This paper positions itself as the first systematic study of expressiveness for several recently‑proposed parametric distributions over DAGs. The authors considered two practical gaps: (i) assign equal mass within a Markov‑equivalence class and (ii) model edge‑dependencies; and introduced (a) a recipe for importance‑sampling evaluation and (b) simple mixture models that overcome the gaps.

**Audience:**

Yes

**Broader Impact Concerns:**

None.

**Claims And Evidence:**

Yes

**Requested Changes:**

Please find my comments and requested changes in Weaknesses above.

**Strengths And Weaknesses:**

**Strengths**

- The topic considered is new and interesting by focusing on expressivity of DAGs
- Provided illustrative counter‑examples (Examples 1 & 2) that showcase (i) even mass across Markov‑equivalent graphs and (ii) edge‑level dependencies.
- Lemma 1 derives closed‑form minima for total variation and KL distances
- Proposed simple fix with empirical gains

**Weaknesses**

- Contribution is essentially a mixture models of distributions. Can you elaborate the novelty over existing literatures of mixture models?
- There lack details for the proposed mixture method, its computational complexities, and its theoretical guarantees. I am also curious how their theoretical gain beyond baseline methods.
- Lemma 1 / the main theory and the proposed method is disconnected. Sections 4.1 and 4.2 are also disconnected. Can you elaborate the   utility of results of Lemma 1?
- The proposed importance-sampling for evaluating implicit models seems very straightforward and lack analysis of variance. Any assumption or condition for the denominator, i.e., the proposal distribution? What if the denominator goes to 0?
- Assumptions need to be more clearly stated, not blended in the preliminaries of "causal structure learning". Do you have any other assumptions? What's "assumed ideal conditions" in Section 5?
- Related work on universal or highly expressive DAG priors—such as DAG-VAE or flow-matching models—is not discussed.
- While the paper includes results for a few mixture sizes (e.g., 2, 5, 10), it lacks systematic ablation or guidelines on how to choose $K$.
- Real‑world utility remain untested. Can the author consider some real applications? Again, it is a bit abstract on how your work can be "helpful to researchers and practitioners alike by demonstrating shortcomings of recently proposed distributions over DAGs". Can you try to assess on downstream causal‑discovery accuracy or decision‑making impact?
- The simulation studies seem to be limited to $D\leq 4$. Can you try a larger scale?
- Hyper‑parameter tuning procedure is under‑specified.
- Several parts of the manuscript are hindered by grammatical errors and unclear phrasing. Figures could be better annotated, and the paper would benefit from substantial proofreading. E.g., " (Example 1 or capture dependences of edgesin the graphs" Please fix them.

---

> ### Author Response · Authors · 2025-08-11
> **Rebuttal 1/3**
>
> *We thank the reviewer for their thoughtful comments and questions. Please find below our responses following the ordering in the review:*
>
> > Contribution is essentially a mixture models of distributions. Can you elaborate the novelty over existing literatures of mixture models?
>
> 1) We have a dedicated paragraph in section 1 on our contributions that are not limited to mixture distributions for causal discovery.
> However, we do show that in our experimental settings mixture models can overcome the limitations of base models (DPM- and ARCO-DAG) and outperform theoretically more expressive models (GFlowNet-DAG) as well as particle distributions with a high number of modeled graphs in practical settings. Our work sheds light on theoretical limitations of distributions over DAGs that were model parts of research papers at top-tier ML conferences and studies an underinvestigated problem.
> To the best of our knowledge, mixture distributions have not been used in causal discovery to model the uncertainty over the prediction and overcome limited expressivity of base models.
>
> > There lack details for the proposed mixture method, its computational complexities, and its theoretical guarantees. I am also curious how their theoretical gain beyond baseline methods.
>
> 2) All covered candidate models are sufficiently flexible to model any (single) DAG, but are only approximations to a true distribution as outlined at the beginning of section 3.
> Trivially, a mixture of $K$ models is strictly more expressive than the same single base model or a distribution with $K$ particles/graphs.
> For the particle distributions we explicitly derive an analytic formula for the statistical distances and divergences in dependence of the number of modeled graphs/particles $K$ in Hence, the optimal values in lemma 1 are upper bounds for the mixture models. In our experiments, these theoretical upper bounds are easily outperformed (except in experiment 1).
> The complexity of the models is hinted by the number of learnable parameters listed in Table 1. Using the score function gradient estimation, each model of the mixture distribution can be evaluated in parallel (given sufficient computational power, yielding the same run time for a single optimization step. Alternatively, a subset of $k<K$  models can be drawn at random and used for training and evaluation.
>
> > Lemma 1/the main theory and the proposed method is disconnected. Sections 4.1 and 4.2 are also disconnected. Can you elaborate the utility of results of Lemma 1?
>
> 3) For section 3, we decided against presenting the limitations of several candidate models as lemmata, but provided simple counterexamples that are straight-forward to understand.
> Sections 4.1 and 4.2 are separate in a sense that they cover the evaluation for different models, but both outline how the models are fitted in the supervised setting in the subsequent experimental section
> Lemma 1 states the optimal values for particle representations and therefore the theoretical limits for a given number of particles. The result is used in the experimental section for the optimal values in Table 2 and 4.
>
> > The proposed importance-sampling for evaluating implicit models seems very straightforward and lack analysis of variance. Any assumption or condition for the denominator, i.e., the proposal distribution? What if the denominator goes to 0?
>
> 4) *On positivity of the denominator:*\
> All considered models (except the particle distributions) assign all DAGs some positive probability mass. Since the proposal distribution is derived from the candidate model, the evaluated probabilities are in the interval (0,1]. Note that we do not evaluate any graphs that are not acyclic.\
> \
> *On the analysis of variance:*\
> The importance-sampling (IS) based on Eq. 15, guarantees that only samples generating the target graph $(\mathbf{G}^\star, \mathbf{Y}^\star)$ are sampled and evaluated to estimate $q_\phi(\mathbf{G}^\star)$. In contrast to direct sampling, it is sample-efficient.
> We followed your suggestion and added a formal analysis of the variance for the proposed IS. In section 4.2 we present our result:
>     > **Lemma 2:** (Uniform sample-efficiency of the IS estimator)\
>     > The effective sample size of the importance-sample estimator $q_\text{IS}$ derived from Eq. 15 is greater than $N$ for all $\mathbf{G}^\star \in \mathcal{G}$ and increases with decreasing values of $q_\phi(\mathbf{G}^\star)$.
>
>     We extended the Appendix “Importance sampling for parametrized distributions with an auxiliary discrete structure” by the proof of lemma 2. For brevity, we provide here only the variances of the standard estimator $q_\text{MC}$ and of the IS estimator $q_\text{IS}$:
>    > $N \text{Var} \big[ q_\text{MC}(\mathbf{G}^\star) \big] = q_\phi(\mathbf{G}^\star) - q_\phi(\mathbf{G}^\star)^2$\
>    > $N \text{Var} \big[ q_\text{IS}(\mathbf{G}^\star)  \big] =  \mathbb{E} \left[
>          r_\phi(\mathbf{G}^*, \mathbf{Y}^\star)^2 \right] - q_\phi(\mathbf{G}^\star)^2$

---

> > ### Author Response · Authors · 2025-08-11
> > **Rebuttal 2/3**
> >
> > > Assumptions need to be more clearly stated, not blended in the preliminaries of "causal structure learning". Do you have any other assumptions? What's "assumed ideal conditions" in Section 5?
> >
> > 5) *On assumed ideal conditions:*\
> > We focus on the expressivity of candidate distribution used in Bayesian causal discovery, therefore we shielded the influence of other error sources such as the uncertainty over parameters and considered a supervised setting (cf. added Definition 1 [Expressivity]). The supervised training with samples is clearly an idealized scenario that we openly emphasize by this very statement. Importantly it does not refer to any assumptions that have not been discussed.\
> > \
> > *On stating assumptions in section 2 “Preliminaries”:*\
> > We transparently communicate and explain all assumptions in the main text, but considered your feedback in the revision of our paper. To distinguish the supervised setting for testing the expressiveness of candidate distributions from the typical unsupervised task of causal discovery, we introduced the general assumptions of the latter already  in section 2.
> >
> >
> > > Related work on universal or highly expressive DAG priors—such as DAG-VAE or flow-matching models—is not discussed.
> >
> > 6) We assume that DAG-VAE refers to the work by Zhang et al. (2019). They do not model a distribution over DAGs, but consider DAGs as the input to their models and encode them in a latent space.
> > To the best of the author’s knowledge and without any provided reference, there is no flow-matching model that models a distribution over DAGs. However, we are happy to add a discussion on such work if you point us to it.
> >
> >     Zhang, M., Jiang, S., Cui, Z., Garnett, R., & Chen, Y. "D-VAE: A variational autoencoder for directed acyclic graphs." Advances in neural information processing systems 32 (2019).
> >
> >
> > > While the paper includes results for a few mixture sizes (e.g., 2, 5, 10), it lacks systematic ablation or guidelines on how to choose K.
> >
> > 7) Please note that we do not state nor claim a method for picking $K$ to handle the trade-off between expressivity and computational complexity.
> > Bayesian causal discovery is an unsupervised task. Therefore, in practice we cannot use our results from the supervised setting for picking an optimal $K$.
> > In principle, it could be chosen using cross-validation, but this is not the scope of our work.
> > Our results show clear limitations of proposed semi-implicit distributions and demonstrate that in our experiments a moderate number of particles, e.g. $K=10$, can overcome them and are competitive behaviour for simple base models (DPM- or ARCO-DAG) with theoretically more expressive models (GFlowNet-DAG).\
> > For Table 1 and 2, we present particle distributions/mixture models for selected values of $K$. If requested, we can also report results for intermediate $K$ values, higher values than the listed ones do not provide more insights, since the initial limitation has been successfully overcome.
> > Please note that Example 1 & 2  and their results in Table 2 allow for direct comparisons (for the same value of $K$) and can be understood as an ablation study. In particular, DPM- and ARCO-DAG only differ in the different distribution over the permutation.
> >
> >
> > > Real‑world utility remain untested. Can the author consider some real applications? Again, it is a bit abstract on how your work can be "helpful to researchers and practitioners alike by demonstrating shortcomings of recently proposed distributions over DAGs". Can you try to assess on downstream causal‑discovery accuracy or decision‑making impact?
> >
> > 8) We did not claim that we had tested or assessed real-world utility, but showed limitations of recently proposed models published at top-tier ML conference highlighting their importance for the research community.
> > Practitioners who want to apply their methods for Bayesian causal discovery to answer causal queries in a subsequent step, clearly benefit from the knowledge of their theoretical and practical limitations and are well-advised to consider mixture models for the distribution over the casual DAGs.
> >
> >
> > > The simulation studies seem to be limited to D. Can you try a larger scale?
> >
> > 9) The posterior distribution from Bayesian causal discovery is seldom evaluated against the true posterior that is typically intractable for even moderate $D$. A notable exception is Deleu et al. (2022) who computed the true posterior for a linear FCM model with 5 variables (only single graph) and evaluated their GFlowNet-DAG on marginal edge probabilities, but not the joint distribution.
> > Our experiments are carefully designed to confirm the theoretical limitations in a very simple setting, but the tested patterns appear in bigger graphs as subgraphs.

---

> > > ### Author Response · Authors · 2025-08-11
> > > **Rebuttal 3/3**
> > >
> > > > Hyperparameter tuning procedure is under‑specified.
> > >
> > > 10) As stated in Appendix B, we performed a grid-search over the learning rate of the Adam optimizer with decoupled weight decay. We did not do a neural-architecture search for the neural networks of ARCO-DAG or GFlowNet-DAG and used the architecture suggested by their original authors. For the particle methods, we computed the optimal analytical values using Lemma 1.
> > >
> > >
> > > > Several parts of the manuscript are hindered by grammatical errors and unclear phrasing. Figures could be better annotated, and the paper would benefit from substantial proofreading. E.g., " (Example 1 or capture dependences of edgesin the graphs" Please fix them
> > >
> > > 11) We corrected the listed sentence and carefully revised our manuscript taking the feedback of the reviewer into account.
> > > With respect to “unclear phrasing”, we would welcome a clear reference to specific sentences or paragraphs in order to address them.
> > >
> > > *Finally, we like to add that while we concentrated in our experimental section on the total variation distance and the KL divergence, Lemma 1 holds also for the Hellinger and Bhattacharyya distance (Strengths).*

---

### Author Response · Authors · 2025-08-20
**Revised submission**

Based on the received reviews, we carefully revised your research article and incorporated the feedback of the reviewers. For clarity and comprehensibility, we provide here an overview of the main changes (for details we refer the reader to the rebuttal and the uploaded revised article):

**Qualitative changes:**
- We added a formal definition of the expressiveness of a distribution at the beginning of section 3 as suggested by reviewer *G1Wi*.
- Reviewer *sSDn* pointed us to the recent preprint by Thompson et al. (2024), for which we included a discussion in section 3.1 and provide the corresponding probability mass function in Appendix A.2.
- To highlight the implications of the different candidate distributions we integrated a brief discussion for the probability mass functions in Appendix A. In addition, we included a dedicated subsection on general mixture distributions as A.6.
- The feedback by reviewer *jnvY* encouraged us to add a more formal analysis of the sample-efficiency of the proposed importance sampling. This results in Lemma 2 and an extended section in appendix C where we proof the optimality of the proposed estimator and openly discuss its slight limitation to an sample-efficient estimator for the considered order-based models.

- As requested by reviewer *4dVp*, we provide now a qualitative comparison of the training times for the considered models in Appendix D.3 alongside the explicit number of model parameters.

**Structural changes:**
- *Enumeration of the equations:*\
We added Eq. 4 for the discussion of the work by Thompson et al. (2024). To maintain the enumeration of Eq. 5 to 15 from the initial submission, we embedded the old Eq. 1 inside the block text.
- *Ordering of the appendix:*\
To match the ordering of the presentation in main text, we shifted the section on the proof of lemma 1 for the particle distribution from the end of the appendix to its second position.

We believe that these changes, combined with the other minor adjustments, further enhance the presentation of our work, and hope that the reviewers agree with us and appreciate the made changes.

---

### Decision · Action_Editor_EPFc · 2025-09-08

**Recommendation:** Accept with minor revision

**Additional Comments:**

In their final evaluations, several reviewers requested some minor revisions to the presentation:
1. In addition to SHD, please also include F1, AUROC, and Brier scores (see Thompson et al 2024 for examples and explanation on how to sample the generative distribution over DAGs to compute these scores).
2. Fix typos for generative distribution as discussed and make the suggested changes to wording (removal of unnecessary sentences) for clarity.
3. Add discussion of Thompson et al 2024.
4. Designate a clear place in the paper to define the proposed method in its entirety. According to one reviewer:
> The authors added explanations about the mixtures, but they appear in section 3.2 together with the particle representations and in the Appendix. In the authors' words, the explanations appear as follows: "We provide an illustrative example in section 3.2 ... The base models ... are well described in section 3.3 and Appendix A. Their mixtures are introduced at the beginning of section 5 and illustrated in Figure 3d. To enhance the presentation, we added a direct note ... at the end of section 3.4 and the following subsection to Appendix A:" which, I believe, highlights how fragmented the presentation is.

Please address each of these points carefully in the final camera ready submission.

**Audience:**

Yes

**Audience Explanation:**

This is a paper about expressivity of causal DAG models, which should be of interest to at least some individuals in TMLR's audience. Reviewers agree it provides a useful consolidation and review of results that will be useful to other researchers.

**Claims And Evidence:**

Yes

**Claims Explanation:**

Aside from a minor concern regarding the choice of metrics, which should be addressed in the camera ready, all reviewers agree the claims in the paper are supported.

---

> ### Author Response · Authors · 2025-10-08
> **Camera ready version**
>
> *We carefully reworked our submission and, in particular, carefully addressed all requested revisions as also listed by the action editor in his decision. The updates clearly improved the clarity of our work. In the following we outline the changes in the camera-ready version w.r.t. to these four points.*
> ___
> **1. Additional metrics.**
> As noted in our rebuttal to reviewer *sSDn* and in our response to the requested changes, our work focuses on investigating the expressiveness of candidate models for representing distributions over graphs. This expressiveness is appropriately measured using statistical divergences, not metrics like SHD, F1, AUROC, or Brier scores, which require a single known target graph which is not available in our setting. The request to revise our paper in this regard appears to stem from a misunderstanding that our work involves causal discovery of a true underlying causal graph. However, our intent is to study aspects of modeling distributions over DAGs (with causal discovery as the downstream application in mind) while minimizing confounding factors. To clarify this, we have included a formal definition of expressiveness, as suggested by reviewer *G1Wi*, in the camera-ready version of our paper.
> ___
> **2. Typos and changes in wording.**
> We have carefully revised our paper in this regard. In particular, we also carefully rephrased the section on "Bayesian causal structure learning” to improve clarity of our presentation.
> ___
> **3.  Discussion of Thompson et al. (2024).**
> We have included/extended our discussion of Thompson et al. (2024) in the camera-ready version. In particular:
>  - In the second paragraph in section 3.1 (including Eq. 4) we present the generative model proposed by Thompson et al. (2024) and discuss the connection to the other candidate models,
>  - In the last paragraph of section 4.2 we explain why the proposed importance-sampling strategy cannot be applied for their model,
>  - Appendix A.2 provides the probability mass function as well as further details on the projection used by Thompson et al.’s generative model.
> ___
> **4. Presentation of the proposed method.**
> As requested, we have designated a clear place for the presentation of the mixture models. To this end, we have split section 3.2 “Particle representation & mixture models” into two separate sections. Now section 3.3 exclusively discusses the idea of mixture models and introduces the general generative model of mixture models and motivates the need for order-based distributions over DAGs as the components of the mixture distribution such as RPM- and ARCO-DAG. Additionally, we provide in appendix A.5 the probability mass function of general mixture models.
> ___